# Structural insights into isoform-specific RAS-PI3Kα interactions and the role of RAS in PI3Kα activation

Daniel Czyzyk [1,6], Wupeng Yan[1,5,6], Simon Messing [1], William Gillette[1], Takashi Tsuji[2], Mitsuhiro Yamaguchi[2], Shinji Furuzono[3], David M. Turner[1], Dominic Esposito [1], Dwight V. Nissley [1], Frank McCormick [1,4] & Dhirendra K. Simanshu [1] ✉

Mutations in RAS and PI3Kα are major drivers of human cancer. Their interaction plays a crucial role in activating PI3Kα and amplifying the PI3K-AKT-mTOR pathway. Disrupting RAS-PI3Kα interaction enhances survival in lung and skin cancer models and reduces tumor growth and angiogenesis, although the structural details of this interaction remain unclear. Here, we present structures of KRAS, RRAS2, and MRAS bound to the catalytic subunit (p110α) of PI3Kα, elucidating the interaction interfaces and local conformational changes upon complex formation. Structural and mutational analyses highlighted key residues in RAS and PI3Kα impacting binding affinity and revealed isoform-specific differences at the interaction interface in RAS and PI3K isoforms, providing a rationale for their differential affinities. Notably, in the RAS-p110α complex structures, RAS interaction with p110α is limited to the RAS-binding domain and does not involve the kinase domain. This study underscores the pivotal role of the RAS-PI3Kα interaction in PI3Kα activation and provides a blueprint for designing PI3Kα isoform-specific inhibitors to disrupt this interaction.

The RAS-driven activation of phosphatidylinositol 3-kinases (PI3K) amplifies the PI3K-AKT-mTOR signaling pathway, a key driver of oncogenic transformation[1–4]. The *PIK3CA* (encodes p110α, the catalytic subunit of PI3Kα) and *RAS* genes rank among the most frequently mutated in human cancers[5]. RAS proteins cycle between inactive GDP-bound and active GTP-bound states. Structural studies have identified switch-I and -II regions that undergo conformational changes during this transition, enabling high-affinity binding to downstream effectors and promoting their activation, leading to downstream signaling[6]. The RAF and PI3K were the first major RAS effectors identified three decades ago. Upon RAS-mediated

activation, RAF and PI3K drive cell growth, proliferation, and survival through the RAF/MEK/ERK and PI3K/AKT/mTOR pathways, respectively. RAF kinases display selective and high-affinity interactions with classical H/N/KRAS proteins and minimal interactions with other RAS family GTPases. In contrast, PI3Ks and other effectors demonstrate a more moderate affinity with classical H/N/KRAS proteins, suggesting a less restricted specificity for RAS family GTPases[7]. Unlike its role as a principal activator of RAF kinases, RAS primarily amplifies PI3K activation, with RAS-mediated PI3K activation playing a less critical role in normal cellular regulation but becoming more prominent in cancer[8].

[1]NCI RAS Initiative, Cancer Research Technology Program, Frederick National Laboratory for Cancer Research, Frederick, MD, USA. [2]Medicinal Chemistry Research Laboratories, Daiichi Sankyo Co, Ltd, Tokyo, Japan. [3]Cardiovascular Metabolic Research Laboratories, Daiichi Sankyo Co., Ltd., Tokyo, Japan. [4]Helen Diller Family Comprehensive Cancer Center, University of California San Francisco, 1450 3rd Street, San Francisco, CA, USA. [5]Present address: School of Life Sciences and Biotechnology, Shanghai Jiao Tong University, Shanghai 200240, China. [6]These authors contributed equally: Daniel Czyzyk, Wupeng Yan. ✉e-mail: dhirendra.simanshu@nih.gov

Previous studies show that, in addition to classical RAS proteins, other RAS family GTPases such as RRAS, RRAS2, MRAS, and ERAS also interact with PI3Kα, indicating their potential role as physiological activators of the enzyme[1,9,10]. The RRAS subfamily, where "R" denotes "Related," comprising RRAS, RRAS2 (TC21), and MRAS (RRAS3) share ~50% amino acid identity with the classical RAS proteins. RRAS, RRAS2, and MRAS GTPases are larger than classical RAS proteins and contain additional N-terminal residues: 26 in RRAS, 11 in RRAS2, and 10 in MRAS. Like classical RAS proteins, RRAS subfamily members possess transforming activities, albeit weakly[11–14]. Mutations in RRAS subfamily genes occur in human cancers at low frequency, and gain-of-function mutations in Noonan syndrome, a RASopathy[15,16]. Though less understood, their physiological roles include regulation of cell morphology, adhesion, and migration[17]. MRAS in complex with SHOC2 and protein phosphatase 1 is involved in ERK pathway regulation[18]. Mutations in RRAS2 are implicated in breast tumorigenesis and metastasis[19], while elevated levels of wild-type RRAS2 are found in various cancers, where RRAS2 has been suggested to play a role in immunological development and homeostasis via the PI3K pathway[17,20].

PI3Ks are classified into three classes (I, II, III) based on structure, tissue distribution, and substrate specificity[21]. Class I PI3Ks, linked with several cancers, comprise a catalytic subunit and a regulatory subunit, further divided into Class IA (p110α, p110β, or p110δ with regulatory subunits p85α, p55α, p50α, p85β, or p55γ) and Class IB (p110γ with regulatory subunits p84 or p101)[22]. Previous studies have suggested that RAS proteins bind and activate p110α, p110γ, and p110δ, whereas p110β is activated by RAC1 and CDC42 from the RHO family[1,9,23]. Active RAS interaction with p110α, along with phosphorylated receptor tyrosine kinases (RTK) interaction with p85, activates PI3Kα, allowing it to modify its lipid substrate, phosphatidylinositol-4,5-bisphosphate (PIP2), to phosphatidylinositol 3,4,5-trisphosphate (PIP3)[21,24,25]. PIP3 recruits proteins with PH domains, such as AKT and PDK1, to the membrane, where PDK1 partially activates AKT by phosphorylating T308, and mTORC2 fully activates AKT by phosphorylating S473. Fully activated AKT phosphorylates numerous downstream substrates, regulating apoptosis and cell survival.

Despite the critical role of the RAS-PI3Kα interaction in PI3Kα activation and PI3K-AKT-mTOR signaling pathway, its structural details have remained elusive. Previous studies have demonstrated that introducing two germline mutations (T208D/K227A) in the RAS-binding domain (RBD) of PI3Kα, which disrupt its interaction with RAS, rendered it resistant to oncogenic RAS-driven lung and skin tumorigenesis in both mice and mouse embryonic fibroblasts[8]. Additionally, disrupting this crucial interaction diminishes tumor growth, metastasis, and tumor-induced angiogenesis[26,27]. This interaction is also pivotal for tumor progression in cancers with WT-RAS, as evidenced by EGFR-mutant lung adenocarcinoma mice harboring PI3Kα-T208D/K227A mutations, where disrupting this interaction inhibits tumor onset and promotes significant tumor regression[28]. Moreover, inhibiting the interaction between PI3Kα and RAS in wild-type mice was observed to be well tolerated[8], underscoring the importance of targeting the p110α-RAS interaction as a potential strategy for cancer treatment. Structural and biophysical studies on the apo-form of wild-type and oncogenic mutants of p110α, with and without p85, have provided critical insights into how p85 regulates the catalytic activity of p110α and the mechanism for mutational activation[29–34]. However, without a structural understanding of the RAS-PI3Kα complex, the detailed interaction interface and impact of PI3Kα-T208D/K227A mutations on the RAS-PI3Kα interaction remains unclear. Although structural data on the HRAS-PI3Kγ complex have been reported[35], the low sequence identity (18% in RBD and 30% overall) and multiple insertions in PI3Kα near the RAS-binding region complicate extrapolation of known RAS-PI3Kγ binding determinants to RAS-PI3Kα.

In this study, we determine the binding affinities of various RAS family GTPases with class I PI3K isoforms (α, β, γ, and δ), revealing distinct affinities and specificities. The binding data shows that RRAS2 and MRAS exhibit a higher affinity for PI3Kα than classical H/N/KRAS proteins. Here, we describe the structures of RRAS2, MRAS, and KRAS in complex with p110α, with the KRAS-p110α complex stabilized by a glue compound, providing detailed insights into the RAS-PI3Kα interaction interface. Sequence and structural comparison among RAS family GTPases provide a rationale for their differential affinity to p110α, while comparative structural analysis of RAS and PI3Kα in apo and complex forms provides insights into RAS-mediated PI3Kα activation. Additionally, structural analysis of the RAS-p110α complex with other class I PI3K isoforms explains their differential affinity and specificity towards RAS GTPases, providing insights for developing PI3Kα-specific inhibitors targeting the RAS-PI3Kα interface with significant therapeutic implications.

## Results
### Binding affinities of RAS subfamily members with PI3Kα and other PI3K isoforms
Previous studies showed that, among the 35 RAS family GTPases, classical H/N/KRAS proteins and their relatives RRAS, RRAS2, MRAS, and ERAS interact with and activate PI3Kα[1,9,10]. To quantify these interactions, we expressed and purified recombinant RAS family GTPases and class I PI3K isoforms and measured their binding affinities using isothermal titration calorimetry (ITC) under physiological salt and pH conditions (Fig. 1A). RRAS2 and MRAS exhibited the strongest binding affinity to PI3Kα, with dissociation constants ($K_D$) of 3.9 and 5.3 μM, respectively. In contrast, classical RAS proteins H/N/KRAS and RRAS exhibited weaker binding affinity to PI3Kα, with $K_D$ values ranging from 17–28 μM (Fig. 1B). ERAS lacked sufficient solubility for ITC experiments, and our inclusion of RIT1 corroborated previous predictions of no binding to PI3Kα[1,9] (Supplementary Fig. 1A). Interestingly, reducing the salt concentration from the physiological 150 mM NaCl to 50 mM NaCl enhanced KRAS binding to PI3Kα, with the $K_D$ decreasing from 16.9 μM to 4.5 μM (Supplementary Fig. 1B). This aligns with previously reported $K_D$ values under low-salt conditions[36], emphasizing the sensitivity of RAS-PI3K interactions to ionic strength. Thermodynamic data from ITC experiments reveal notable differences in the interactions between RAS proteins and PI3Kα. RRAS2 and MRAS show a ΔH of −17 kcal/mol and a −TΔS of 9 kcal/mol, indicating a strong and entropically favorable interaction (Supplementary Table 1). In contrast, H/N/KRAS display ΔH values ranging from −3 to −8 kcal/mol and −TΔS values from −3.4 to 2.1 kcal/mol, suggesting weaker and more ordered interactions with PI3Kα. Comparable binding affinities of these RAS GTPases to PI3Kα-RBD suggest its vital role in the RAS-PI3Kα interaction (Supplementary Fig. 2). We next evaluated the binding affinities of RRAS2, the strongest PI3Kα interactor, and KRAS, representing classical RAS proteins, with other class I PI3K isoforms: PI3Kβ, PI3Kγ, and PI3Kδ (Fig. 1C and Supplementary Fig. 3). Consistent with previous studies, neither RRAS2 nor KRAS interacted with PI3Kβ. Both GTPases showed similar differential affinities for PI3Kα and PI3Kγ, while their affinities for PI3Kδ were comparable, with a $K_D$ of 14 μM (Fig. 1D). These observations suggest that divergent evolutionary paths have shaped the RBD sequences and structures of PI3K isoforms and RAS family GTPases, resulting in varied affinities and specificities among them.

### The overall structure of RAS-p110α complexes
To elucidate the structural basis of RAS-PI3Kα interactions and the determinants governing affinity and specificity, we attempted to crystallize PI3Kα or p110α in complex with RAS GTPases RRAS2, MRAS and KRAS (Fig. 2A). We successfully isolated stable complexes of RRAS2 and MRAS with PI3Kα using gel-filtration chromatography, reflecting their higher affinity. Attempts to crystallize RRAS2/MRAS with PI3Kα (comprising p110α [2-1068] and p85 [332-600]) were unsuccessful; however, p110α (105-1068), lacking the adapter binding

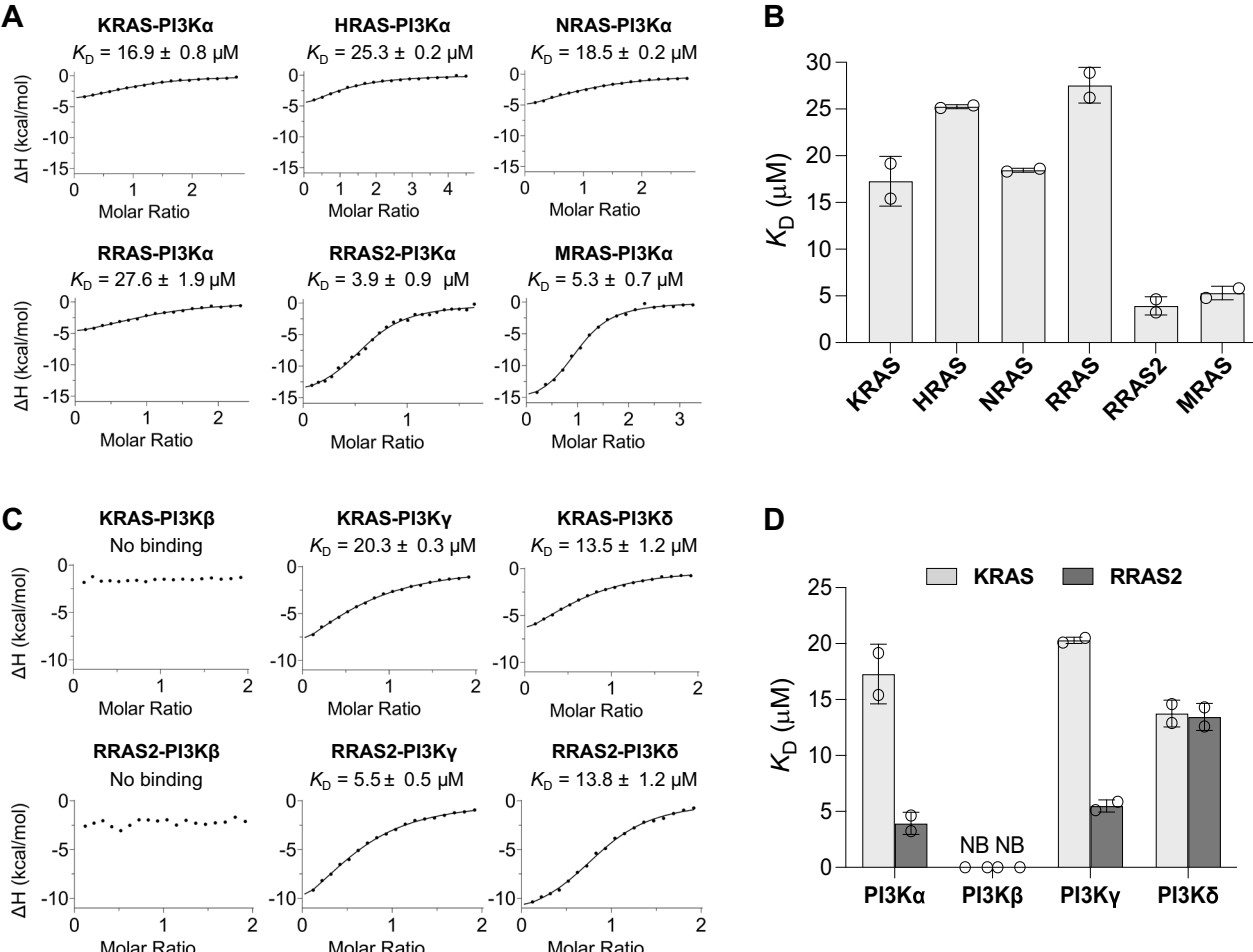

**Fig. 1 | Comparative analysis of the binding affinity of RAS family members with PI3Kα and other class I PI3K isoforms. A** Representative ITC measurements showing the binding affinity of PI3Kα with RAS family members KRAS, HRAS, NRAS, RRAS, RRAS2, and MRAS. $K_D$ values are calculated from two technical replicates (**B**) Bar graph comparing the binding affinity ($K_D$) calculated from two technical replicates of PI3Kα with RAS family members shown in panel (**A**). The error bars define the data range, with two replicates shown as circles. **C** Representative ITC measurements show the binding affinity of KRAS and RRAS2 with three other class I PI3K isoforms: PI3Kβ, PI3Kγ, and PI3Kδ. $K_D$ values calculated from two technical replicates (**D**) Bar graph comparing the binding affinity ($K_D$) calculated from two technical replicates of KRAS (light gray) and RRAS2 (dark gray) with the four class I PI3K isoforms shown in panels **A** and **C**. Error bars define the range of the data, with two replicates displayed as circles. NB: no binding.

domain (ABD; 1-104) that binds p85, produced crystals that diffracted to 4-4.5 Å. Previously, the V223K mutation in PI3Kγ, which enhances binding affinity to RAS, was used to improve crystal quality[35]. However, the equivalent V193K mutation in p110α did not exhibit a similar enhancement in RAS binding affinity (Supplementary Fig. 4A). Recent findings, including our own, demonstrated that Q25A and Q25L mutations in KRAS increase its affinity for effector proteins[37,38]. We observed that the Q25A mutation in KRAS indeed enhanced its binding to PI3Kα compared to wild-type KRAS (Fig. 2B and Supplementary Fig. 4B). Given the conserved glutamine residue in KRAS, RRAS2, and MRAS, we introduced analogous mutations (RRAS2-Q36A and MRAS-Q35A) and assessed their binding affinity to p110α. The RRAS2-Q36A mutant exhibited a twofold increase in binding affinity for PI3Kα, while the MRAS-Q35A mutant displayed comparable affinity to wild-type MRAS (Fig. 2B and Supplementary Fig. 4B). Crystallization of KRAS-Q25A, RRAS2-Q36A, and MRAS-Q35A with p110α (105-1068) resulted in crystals of RRAS2-p110α and MRAS-p110α complexes, enabling structure determination at 3.1 Å and 2.75 Å, respectively (Fig. 2C, D and Table 1). Despite similar p110α binding affinities, the MRAS-Q36A mutant formed better-diffracting crystals than WT-MRAS, possibly due to reduced conformational flexibility or altered surface properties.

We recently screened a small-molecule library for anti-diabetic compounds that mimic insulin's action and identified compounds that bind to p110α as molecular glues, enhancing its interaction with RAS[39]. The glue compound D927 increased the KRAS-PI3Kα interaction affinity by three orders of magnitude, facilitating the stable complex formation and enabling structure determination of the KRAS-p110α complex at 2.81 Å resolution (Fig. 2E, F; Table 1 and Supplementary Fig. 5A, B).

Crystal structures of RAS-p110α complexes reveal well-defined density for the RBD, C2, helical, and kinase domains of p110α, as well as the G-domain of RRAS2, MRAS, and KRAS, with both proteins similarly arranged (Fig. 2C, D, F and Supplementary Fig. 6A, B). In the KRAS-p110α complex, the glue D927 is positioned near the interface, primarily binding to p110α and interacting with KRAS residues (Fig. 2F and Supplementary Fig. 5C, D). All three structures show well-defined density for p110α residues 233–246 in the RBD, forming the rα1-rα2 loop and rα2 helix ("r" for RBD) (Supplementary Fig. 7). This region, disordered in previously reported apo-PI3Kα structures lacking crystal contacts, is stabilized by the PI3Kα-RAS interaction. The rα1-rα2 loop and rα2 helix adopt distinct conformations in the p110α complexes with different RAS GTPases, driven by residue variations Y38 in RRAS2, and its equivalents, I37 in MRAS and H27 in KRAS. These RAS residues

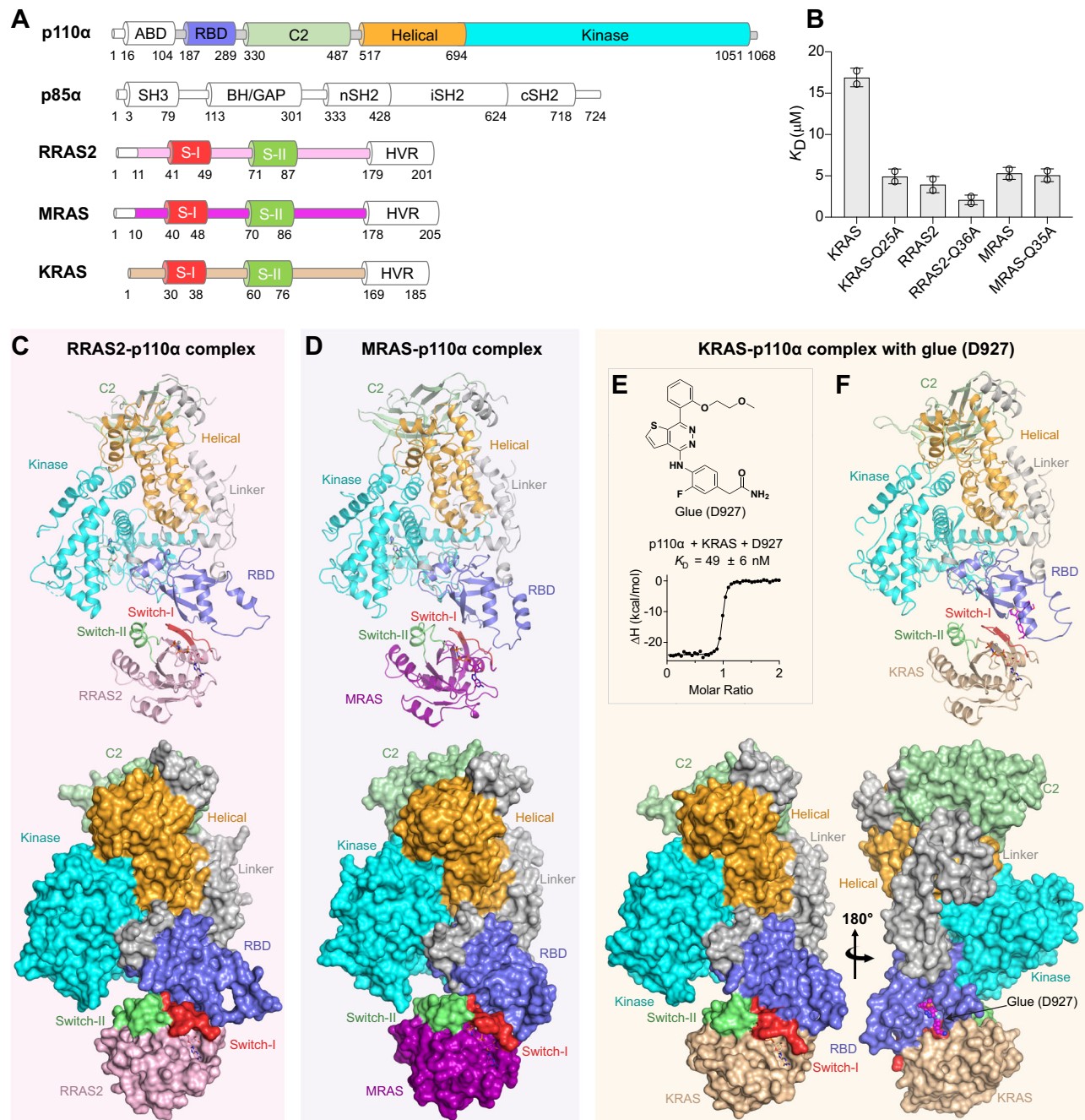

**Fig. 2 | Domain architecture and crystal structures of RRAS2, MRAS, and KRAS in complex with p110α. A** Domain architecture for the catalytic (p110α) and regulatory (p85α) subunits of PI3Kα, alongside RAS GTPases used in this study. Domains or subunits shown in white were not included in the crystallization constructs utilized in this study. **B** Bar graph comparing the binding affinities calculated from two technical replicates of PI3Kα-RBD with wild-type and point mutants of KRAS, RRAS2, and MRAS, showing that the mutants of KRAS and RRAS2 exhibit enhanced affinity for forming stable complexes with p110α. The data range is shown as error bars, with two replicates shown as circles. **C** Overall structure of the RRAS2-p110α complex shown in cartoon and surface representations. **D** Overall structure of the MRAS-p110α complex depicted in cartoon and surface representations. **E** Chemical structure of the glue compound D927 and an ITC profile

showing a significant increase in the binding affinity between KRAS (GMPPNP) and p110α-RBD in the presence of D927. $K_D$ values are presented as the mean ± standard deviation derived from two replicates. **F** Overall structure of the KRAS-p110α complex in the presence of glue D927 is shown in cartoon and surface representations. The p110α RBD, C2, helical, and kinase domains are colored blue, pale green, orange, and cyan, respectively, and the linker regions between these domains are shown in light gray. The ABD domain, depicted in white in panel A, was excluded from the crystallization constructs and is therefore absent in panels (**C–F**). RRAS2, MRAS, and KRAS are colored pink, purple, and wheat, respectively, with the switch-I and switch-II regions highlighted in red and green, respectively. The nucleotide GMPPNP is depicted as sticks, glue compound D927 is shown as magenta-colored sticks or spheres, and Mg²⁺ ions are represented as gray spheres.

interact differently with S231 and M232 in the p110α-RBD rα1-rα2 loop (Supplementary Fig. 8A), highlighting RAS isoform-specific interactions and conformations. While the switch-I region and β2-strand are closely aligned across the RAS proteins, these specific interactions cause KRAS to rotate by ~3 degrees and MRAS by ~6 degrees relative to

RRAS2, emphasizing the structural adaptations from isoform-specific RAS-p110α interactions.

Structural analysis of the RRAS2-p110α complex indicates that the large, polar side chain of glutamine at position 36 in RRAS2 likely disrupts interactions with the rα1-helix residues (S231 and M232) of

**Table 1 | Crystallographic data collection and refinement statistics**

|  | RRAS2-Q36A (GMPPNP) + p110α complex | MRAS-Q35A (GMPPNP) + p110α complex | KRAS (GMPPNP) + p110α + D927 (glue) complex | RRAS2 (GMPPNP) | MRAS (GMPPNP) |
|---|---|---|---|---|---|
| PDB ID | 9B4S | 9B4T | 9C15 | 9B4Q | 9B4R |
| **Data collection** |  |  |  |  |  |
| Space group | $P4_22_12$ | $P2_1$ | $P2_1$ | $P2_12_12_1$ | $P4_22_12$ |
| Cell dimensions |  |  |  |  |  |
| a, b, c (Å) | 181.62, 181.62, 96.21 | 58.87, 105.75, 114.40 | 58.37, 126.64, 113.08 | 39.11, 62.48, 65.69 | 65.56, 65.56, 107.92 |
| α, β, γ (°) | 90.0, 90.0, 90.0 | 90.0, 97.46, 90.0 | 90.0, 104.76, 90.0 | 90.0, 90.0, 90.0 | 90.0, 90.0, 90.0 |
| Resolution (Å) | 50.00–3.10 (3.29 – 3.10)* | 50.00–2.75 (2.92 – 2.75)* | 50.00–2.81 (2.99 – 2.81)* | 50.00–1.46 (1.54 – 1.46)* | 50.00–2.10 (2.22 – 2.10)* |
| $R_{merge}$ | 0.067 (2.038) | 0.078 (1.119) | 0.093 (0.771) | 0.083 (1.230) | 0.010 (0.075) |
| $R_{pim}$ | 0.031 (0.803) | 0.048 (0.656) | 0.081 (0.659) | 0.036 (0.571) | 0.049 (0.452) |
| $I/\sigma I$ | 19.12 (1.27) | 13.48 (1.17) | 10.85 (1.83) | 15.46 (1.71) | 15.17 (2.30) |
| Completeness (%) | 94.8 (96.3) | 98.7 (97.7) | 98.1 (92.3) | 99.8 (98.9) | 99.2 (97.8) |
| Redundancy | 9.3 (9.4) | 7.0 (7.2) | 3.3 (3.1) | 12.5 (12.6) | 8.0 (7.0) |
| $CC_{1/2}$ | 99.9 (43.5) | 99.9 (87.0) | 99.6 (74.9) | 99.9 (90.4) | 99.9 (76.8) |
| **Refinement** |  |  |  |  |  |
| Resolution (Å) | 49.26 – 3.10 | 50.00–2.75 | 45.54 – 2.81 | 45.52 – 1.46 | 42.54 – 2.10 |
| No. reflections | 261,603 | 252,519 | 126,910 | 367,262 | 114,778 |
| $R_{work}$ / $R_{free}$ | 23.2/27.2 | 21.5/26.2 | 20.3/23.4 | 18.4/21.6 | 20.2/25.0 |
| No. atoms |  |  |  |  |  |
| Protein | 8524 | 8626 | 8345 | 1418 | 1316 |
| Ligand/ion | 61 | 61 | 74 | 33 | 35 |
| Water |  | 27 | 38 | 175 | 170 |
| B factors |  |  |  |  |  |
| Protein | 150.4 | 114.6 | 99.6 | 29.2 | 37.3 |
| Ligand/ions | 122.5 | 95.6 | 92.2 | 21.2 | 36.1 |
| Water |  | 93.8 | 63.2 | 39.4 | 45.1 |
| R.m.s. deviations |  |  |  |  |  |
| Bond length (Å) | 0.001 | 0.002 | 0.003 | 0.012 | 0.003 |
| Bond angles (°) | 0.385 | 0.406 | 0.653 | 1.316 | 0.533 |
| Ramachandran |  |  |  |  |  |
| Favored (%) | 92.75 | 93.41 | 95.49 | 98.19 | 98.75 |
| Allowed (%) | 7.15 | 6.49 | 4.51 | 1.81 | 1.25 |
| Outliers (%) | 0.10 | 0.10 | 0.00 | 0.00 | 0.00 |

*Values in parentheses are for the highest-resolution shell.

p110α-RBD (Supplementary Fig. 8B). Replacing glutamine with the smaller, non-polar alanine alleviates these unfavorable interactions, likely leading to the two-fold increase in binding affinity for RRAS2-Q36A compared to WT-RRAS2. Variation at position Y38 in RRAS2 and the equivalent residues I37 in MRAS and H27 in KRAS induce unique conformational changes in the rα1-rα2 loop and rα2-helix of p110α-RBD. Consequently, mutations of nearby residues RRAS2-Q36A, MRAS-Q35A, and KRAS-Q25A show varied effects on binding affinity with p110α.

**Structural analysis of the RAS-p110α interaction interface**
The interaction interfaces of RRAS2, MRAS, and KRAS with p110α display significant similarity, with key interactions conserved across all three structures. Since RRAS2 binds p110α with the highest affinity among the three RAS GTPases, we used the structure of RRAS2-p110α complex for detailed analysis. Given that MRAS and RRAS2 have 10 and 11 extra N-terminal residues compared to KRAS, adjustments in residue numbering are required to identify equivalent residues for comparison (Supplementary Fig. 6B).

Interactions between RAS and p110α-RBD involve two sets of p110α residues: D203 to K210 and K227 to M232 (Fig. 3A–F). The first

set, comprising D203, K204, Q205, K206, Y207, T208, and K210, located on the rβ2-strand of p110α-RBD, engages with residues on the β2-strand and switch-I of RAS (Fig. 3B, C). Specifically, p110α residues K206 and T208 form H-bond and salt-bridge interactions with RRAS2 residues E48 and S50, respectively. Additionally, D203 and K210 at the ends of this set form salt-bridges with R63 in the interswitch and switch-II regions of RRAS2, respectively (Fig. 3B, D). Notably, as R63 in RRAS2 is not conserved in MRAS (I62) and KRAS (L52), p110α residue D203 engages uniquely across the three structures by forming hydrogen bonds with R63 in RRAS2 and H53 in MRAS, and van der Waals interactions with R41 of KRAS (Fig. 3F and Supplementary Fig. 9A–F). The second set of p110α residues, from K227 to M232, located at the end of the rα1-helix and beginning of the rα1-rα2 loop, interacts with residues within and around switch-I region of RAS proteins (S28-D49 in RRAS2) (Fig. 3C, F). Additionally, switch-II residues F75 and M78 of RRAS2 interact with the RBD, forming long-range interactions (Fig. 3D). Key interactions include K227, which forms salt-bridge and H-bond interactions with D44 and D49 of RRAS2 and H-bond interactions with D33 in KRAS and P44 (main chain) in MRAS (Fig. 3C and Supplementary Fig. 9C). In MRAS and RRAS2 complexes, the guanidinium group of R230 forms salt-bridge and H-bond

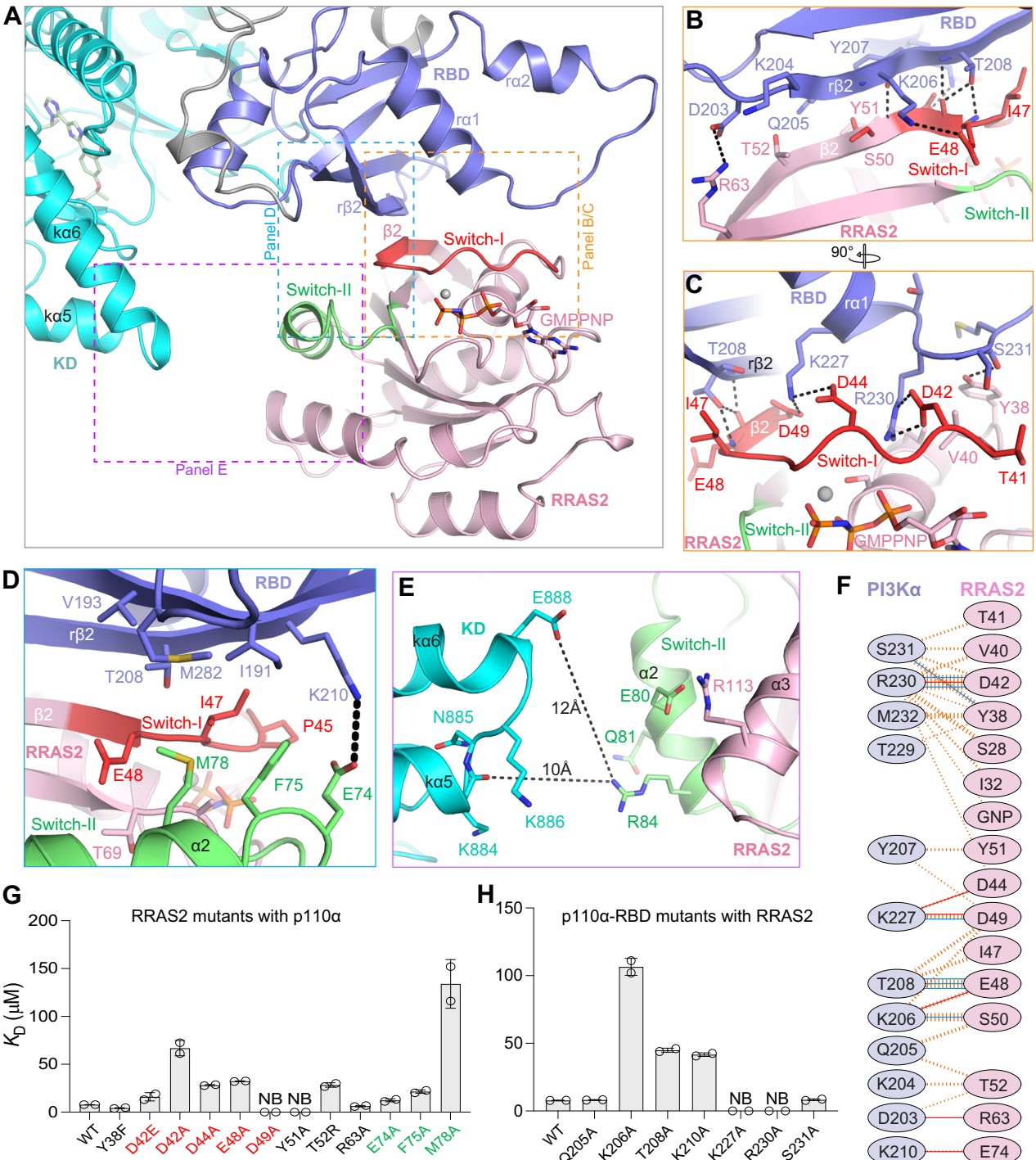

**Fig. 3 | Structural and mutational analysis of the RAS-p110α interaction interface. A** Enlarged view of the RRAS2-p110α complex highlighting the interaction interface, with p110α domains and RRAS2 switch regions color-coded as in Fig. 2A. **B** Interaction between the β2-strand of RRAS2 and the rβ2-strand of p110α. **C** Details of the interactions between the switch-I region (red) of RRAS2 and the rα1-helix and rα1-rα2 loop of the p110α RBD. **D** Interactions of the switch-II region (green) of RRAS2 with the p110α RBD. **E** Interaction interface displaying the switch-II region (green) of RRAS2 and the C-lobe (cyan) of the kinase domain of p110α. The shortest distance between two atoms in these regions, including the distance between E888 of p110α and R84 of RRAS2, is indicated by a dotted line.

**F** Schematic overview of the RRAS2-p110α interaction interface. Interactions are denoted by solid blue lines for hydrogen bonds, solid red lines for salt bridges, and striped-orange lines for non-bonded contacts (the width of the striped line is proportional to the number of atomic contacts). **G** Bar graph showing the binding affinity ($K_D$) calculated from two technical replicates measured using ITC for point mutants of RRAS2 interface residues with p110α. Mutations within the switch-I and switch-II regions are shown in red and green, respectively. **H** Bar graph presenting the binding affinity ($K_D$) calculated from two technical replicates measured using ITC for point mutants of p110α interface residues with RRAS2. The error bar defines the range of the data, with two technical replicates shown as circles. NB: no binding.

interactions with D42 of RRAS2 and D41/D43 of MRAS, respectively (Fig. 3C and Supplementary Fig. 9C, F). The conformation of R230 in the KRAS-p110α complex differs due to a longer side chain in glutamate (E31) in KRAS compared to aspartate in RRAS2 and MRAS. Overall, interactions between RAS and p110α-RBD extend beyond the switch-I and include residues in the interswitch and switch-II regions. Structural analysis shows that RRAS2-p110α and MRAS-p110α complexes have more H-bonds and salt-bridges at the interface than KRAS-p110α complex due to non-conserved interface residues in RAS GTPases, providing a rationale for the higher affinity binding of RRAS2 and MRAS with p110α compared to KRAS.

In all three RAS-p110α structures, the C-lobe of the kinase domain of p110α is positioned near the switch-II region of RAS proteins, with the closest distance ranging from 7.5-10 Å (Fig. 3E and Supplementary Figs. 8C, 9E and 10). In the HRAS-PI3Kγ complex[35], an interaction (4.6 Å) exists between E919 (C-lobe of the kinase domain) of PI3Kγ and R73 (Switch-II) of HRAS. Although these two residues are conserved in p110α and RRAS2/MRAS/KRAS, the distance between them is ~9–12 Å in the structures described here.

### Mutational analysis of the RAS-p110α interaction interface

To discern the contribution of specific interface residues to the RAS-p110α interaction, we mutated key interface residues in RRAS2 and p110α and assessed their binding affinity using ITC (Fig. 3G and Supplementary Fig. 11). Alanine mutations of five switch-I residues in RRAS2—D42, D44, E48, D49, and Y51—significantly reduced its binding affinities to p110α, with mutations D49A and Y51A completely abolishing the interaction. These two RRAS2 residues are conserved in MRAS and KRAS, and mutational data underscore the critical interactions of D49 (RRAS2) with K227 (p110α) and Y51 (RRAS2) with Y207 (p110α). Mutation of D42 in RRAS2 (D41 in MRAS and E31 in KRAS), present in the switch-I region, to glutamate, impacted the RRAS2-p110α-RBD interaction less than the alanine mutation, implying the importance of a salt-bridge interaction between D42 (RRAS2) and R230 (p110α). The Y38 residue of RRAS2, located before switch-I and interacting with M232 and S231 of p110α, showed minimal impact when mutated to phenylalanine, indicating the aromatic ring's role in RAS-p110α interaction. The mutation of T52 in RRAS2 (L51 in MRAS and R41 in KRAS) to arginine resulted in a 7-fold reduction in binding affinity to p110α suggesting compositional differences at this position likely influence RAS GTPase affinity for p110α. The mutation of interswitch residue R63 in RRAS2 (I62 in MRAS and L52 in KRAS), which forms a salt-bridge with D203 (p110α), an interaction only observed in RRAS2-p110α complex, to alanine did not affect the RAS-p110α interaction. Additionally, alanine mutations of switch-II residues E74, F75, and M78 in RRAS2 (conserved in MRAS and KRAS, with F75 equivalent to Y64 in KRAS) reduced affinity for p110α, with the M78A mutation having the most pronounced effect, followed by F75A and then E74A. Overall, our mutational analysis highlighted the significant roles of the D42, D49, and Y51 residues from switch-I and M78 from switch-II in the RRAS2-p110α interaction, with all four residues conserved in RRAS2, MRAS, and KRAS.

To evaluate the role of seven p110α-RBD interface residues, we mutated them to alanine and analyzed their binding affinity to RRAS2 using ITC (Fig. 3H and Supplementary Figs. 9G and 12). K227A and R230A mutants of p110α lost binding to RRAS2, while the K206A mutant showed about a 15-fold reduction in affinity, and T208A and K210A mutants exhibited a 5-fold decrease. These five p110α residues form H-bonds or salt-bridge interactions with RRAS2, MRAS, and KRAS in their complexes with p110α. Mutation of Q205 and S231 to alanine did not affect binding, indicating their minimal role in the complex formation. Our analysis identified critical interactions formed by p110α residues K227 and R230 and highlighted the importance of K206, T208, and K210 in stabilizing the RAS-p110α complex.

### Dissecting the binding affinities differences among RAS family GTPases and p110α

Phylogenetic analysis reveals a distinct cluster of RAS subfamily GTPases that interact with PI3Kα, reflecting their shared evolutionary history and sequence similarity (Supplementary Fig. 13). Sequence and structural differences in p110α-interacting residues between RRAS2/MRAS and classical RAS proteins (H/N/KRAS) likely account for the higher binding affinities of RRAS2 and MRAS for p110α (Fig. 4A). Key non-conserved interface residues in H/N/KRAS—H27, D30, E31, R41, and Y64—are located in or near the switch regions (Fig. 4B–D). Mutating these residues in KRAS to their RRAS2 equivalents – H27Y, D30T, R41T, and Y64 F—slightly improved binding to p110α, while E31D brought KRAS binding affinity to p110α close to RRAS2 and MRAS levels (Fig. 4E and Supplementary Fig. 14). Notably, glutamate (E31) is conserved in H/N/KRAS but diverges to aspartate in RRAS2 (D42) and MRAS (D41), forming a critical salt-bridge with R230 of p110α. The significance of this interaction is underscored by the complete loss of RAS-p110α binding observed with the p110α-R230A mutation. The absence of electron density for the side chain carboxyl group of E31 in KRAS-p110α suggests inherent flexibility. The larger glutamate side chain in KRAS, compared to aspartate in RRAS2 and MRAS, induces a distinct conformation of R230 in KRAS-p110α and MRAS/RRAS2-p110α complexes (Fig. 4F), altering interaction profiles and explaining the higher affinity of RRAS2 and MRAS for p110α compared to H/N/KRAS.

We examined the impact of seven p110α-RBD interface mutants on KRAS-p110α interaction (Fig. 4G–I and Supplementary Fig. 15). These p110α mutants generally displayed interaction profiles with KRAS similar to those observed with RRAS2 (Fig. 3H). Notably, the Q205A mutation reduced the affinity to KRAS by 2-fold while not impacting RRAS2-p110α interaction. Additionally, the K206A mutation completely abolished interaction with KRAS while reducing binding affinity to RRAS2 by 15-fold. Like RRAS2, the K227A and R230A mutations entirely disrupted KRAS binding, whereas the T208A and K210A mutations slightly reduced it (Fig. 4I). The T208D mutation, previously employed in knockout mouse studies[8,28], likely caused steric clash at the interface due to its larger aspartate side chain compared to alanine, along with charge repulsion with KRAS residues E37 and D38, resulting in complete loss of KRAS binding (Fig. 4G).

Furthermore, we evaluated the impact of mutations in KRAS residues I36, D38, S39, and Y40, located on the β2-strand at the KRAS-p110α interface near the glue interacting site (Fig. 4G–K). The results showed that these residues play critical roles in the KRAS-p110α interaction, as mutating them to alanine resulted in either significant loss (S39A) or complete loss (I36A, D38A, Y40A) of the KRAS-p110α interaction (Supplementary Fig. 14). We also investigated the role of KRAS Y64 in the KRAS-PI3Kα interaction. While the Y64F mutation had no effect, the Y64A mutation completely abolished KRAS-p110α binding. Interestingly, the analogous F75A mutation in RRAS2 only minimally impacted its p110α interaction. The Y64A mutation in KRAS likely induces significant structural alterations in the switch-II region due to compositional differences between KRAS and RRAS2, potentially affecting the interactions of neighboring KRAS residues E63 and M67 with p110α (Fig. 4H).

Structural analysis shows that the glue compound D927 binds to the p110α-RBD at the interface within a pocket formed by the central β-sheet and rα1-helix (Fig. 4K and Supplementary Fig. 5D). The p110α residues V196, Q205, Y207, K228, Y246, Y250, and L287 form the D927 binding pocket and predominantly interact via hydrophobic and van der Waals forces. These interactions stabilize the rα1-rα2 helix and intervening loop of p110α-RBD in a conformation conducive to RAS binding. At the KRAS-p110α interface, the amide group of D927 forms an H-bond with R41 (KRAS) and engages in van der Waals interactions with Y40 (KRAS). Thus, D927 substantially enhances p110α's affinity for RAS by forming additional interactions

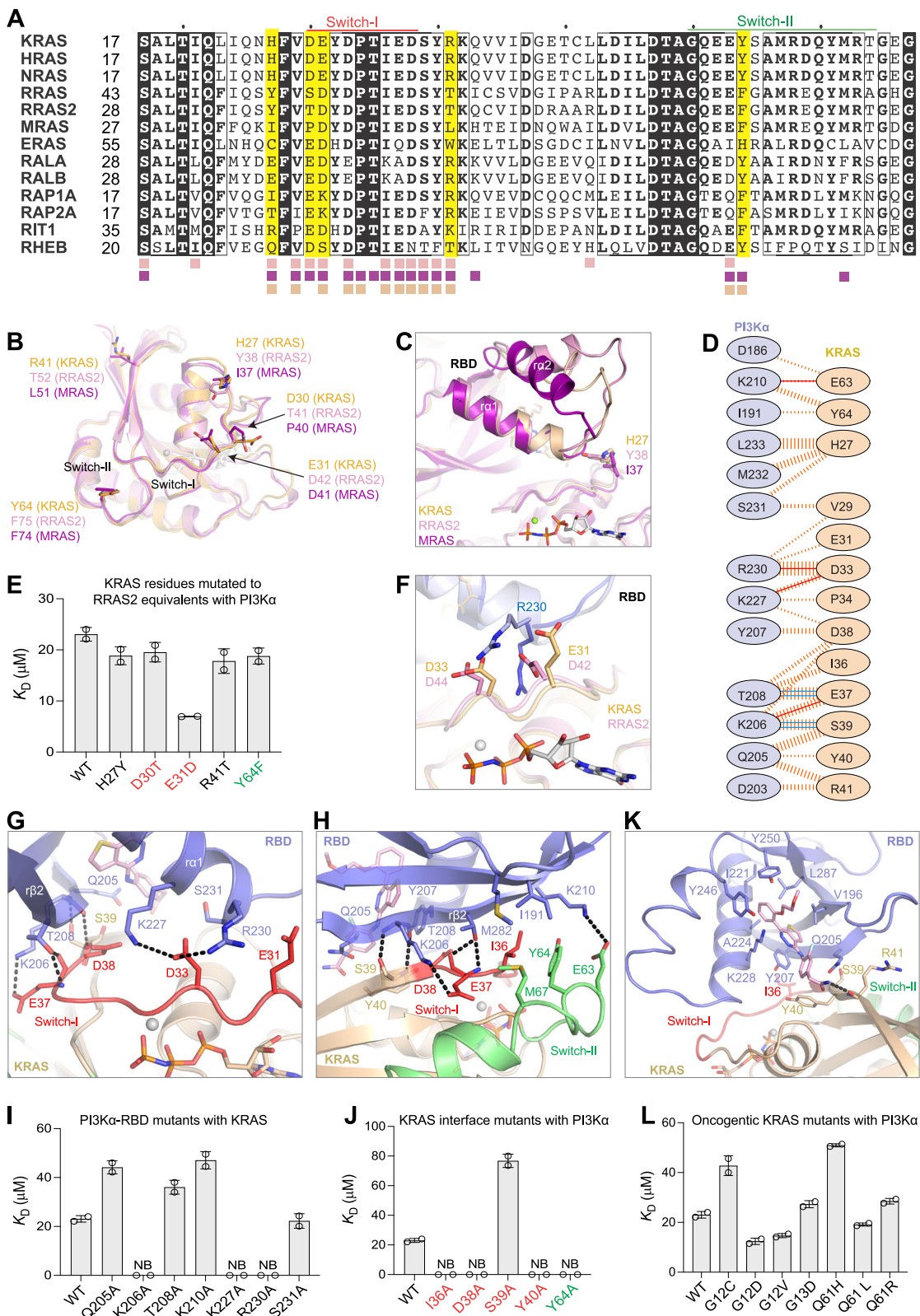

and stabilizing secondary structures containing key interacting residues.

## Oncogenic KRAS and RRAS2 mutations and their effect on p110α binding

We investigated the effects of oncogenic KRAS mutations at G12, G13, and Q61 on their binding with p110α-RBD (Fig. 4L and Supplementary

Fig. 16). The KRAS-G12D mutant displayed approximately a 2-fold increase in binding affinity to p110α-RBD, with a similar increase observed for G12V. Conversely, G12C and Q61H mutants showed a 2-fold decrease in binding affinity. Other mutants, including G13D, Q61L, and Q61R, demonstrated binding affinities with p110α-RBD comparable to WT-KRAS. Though residues at positions 12, 13, and 61 do not directly interact with p110α, local conformational changes

**Fig. 4 | Structural analysis of KRAS, RRAS2, and MRAS reveals key differences driving their differential affinity to p110α. A** Sequence alignment of RAS family GTPases in and around the switch regions. Fully conserved residues are highlighted in black, while similar or predominantly conserved positions are enclosed in boxes. Pink, purple, and orange squares beneath the aligned sequences indicate residues from RRAS2, MRAS, and KRAS, respectively, that interact with p110α. Five key interface residues that differ in KRAS, RRAS2, and MRAS are highlighted in yellow. **B** Overlay of RRAS2 (pink), MRAS (purple), and KRAS (orange) from their respective complexes with p110α, showing key compositional differences in the residues that interact with p110α. **C** Compositional differences in RRAS2, MRAS, and KRAS at the interaction interface result in unique conformational changes in the rα2-helix and rα1-rα2 loop of the p110α-RBD. **D** Schematic overview of the KRAS-p110α interaction interface, using the same notations described for panel 3 F. **E** Bar graph displaying the binding affinity ($K_D$) calculated from two technical replicates of p110α-RBD with KRAS mutants in which a KRAS interface residue was mutated to the

corresponding RRAS2 residue. **F** Structural superposition of the KRAS-p110α and RRAS2-p110α complexes showing conformational changes in the side chain of R230 of p110α. **G** Details of the interactions between the switch-I region of KRAS with the rβ1-strand and rα1-helix of the p110α RBD. **H** Details of the interactions between the switch-II region and the β2-strand of KRAS with the rβ2-strand of the p110α RBD. **I** Bar graph depicting the binding affinity ($K_D$) calculated from two technical replicates of point mutants of p110α RBD with GMPPNP-bound KRAS. **J** Bar graph displaying the binding affinity ($K_D$) calculated from two technical replicates for point mutants of KRAS interacting with p110α-RBD, with mutations in the switch-I and switch-II regions of KRAS highlighted in red and green, respectively. **K** Enlarged view of the glue binding pocket showing p110α and KRAS residues interacting with the glue compound D927. **L** Bar graph showing the binding affinity ($K_D$) of oncogenic mutants of KRAS at positions 12, 13, and 61 with p110α-RBD. The error bars define the data range, with two replicates shown as circles. NB: no binding.

induced by mutations at these sites by different residues could potentially lead to minor alterations at the interface, impacting their binding affinities. These results suggest that oncogenic mutations in KRAS not only impact intracellular RAS-GTP levels but also modulate their binding affinity with PI3Kα.

We also examined RRAS2 mutations (G23D, G23V, Q72H, and Q72L) detected in Noonan syndrome and cancer to assess their impact on affinity to PI3Kα[15,16] (Supplementary Fig. 17). These RRAS2 mutants predominantly exhibited binding affinity to PI3Kα similar to WT-RRAS2, with minor reductions for G23V and Q72H. These results suggest that the clinical manifestations of these RRAS2 mutations likely stem from increased intracellular RRAS2-GTP levels due to impaired GAP-mediated GTP hydrolysis.

### Conformational changes in p110α and RAS upon complex formation

To determine if p110α and RAS undergo conformational changes upon complex formation, we conducted comparative structural analyses using the structure of p110α and RAS GTPases in their uncomplexed states. Structural superposition of apo-p110α[40] (PDB: 8OW2) with its complexes with RAS proteins reveals RMSDs ranging from 1.01–1.39 Å, indicating minimal conformational changes upon RAS binding (Fig. 5A, B). Notable changes are localized to the RBD rα1-rα2 loop and rα2-helix (residues 231–247), the C2 domain (residues 408–420), the C2-helical domain loop (residues 500–529), and the C-terminus (residues 1042–1048) of the kinase domain (Fig. 5A, C). Additionally, subtle alterations are detected at the N-lobe (residues 726–742) and C-lobe (residues 861–874, facing the switch-II region) of the kinase domain, as well as the region following the activation loop (residues 969–972) (Fig. 5A). However, apart from the RBD, these changes occur in regions with inherently high flexibility, posing challenges in discerning whether they stem from RAS binding or intrinsic structural dynamics captured differently in crystal structures. The activation loop is disordered in all three p110α-RAS complexes, as observed in previously reported apo-p110α structures. These findings indicate that RAS binding to p110α causes conformational changes primarily in the RBD, with minimal effects in the kinase and other domains.

Comparative structural analysis of GMPPNP-bound RRAS2 (this study) and KRAS (PDB: 6GOD) in their free forms (without p110α), aligned with their p110α complexes, reveals RMSDs of 0.67 Å and 0.97 Å (Supplementary Fig. 18A-E). Both RRAS2 and KRAS maintain similar switch region conformations in their free and p110α-bound states, indicating that GMPPNP binding induces the effector-binding conformation. Conversely, the structure of GMPPNP-bound MRAS (this study) without p110α lacks electron density for switch-I residues 41–51, indicating increased flexibility and likely adopting a state 1 conformation (RMSD of 0.71 Å, excluding switch I, compared to the MRAS-p110α complex) (Supplementary Fig. 18C). Binding of p110α is likely to shift MRAS switch regions from state 1 to the effector-binding

state 2 conformation. Solution NMR studies have shown that active RAS proteins transition between two conformational states, termed State 1 and State 2. State 1 is characterized by a more 'open' Switch-I region, and State 2 is a 'closed' conformation associated with effector interactions[41]. Superposition of the KRAS-GDP (PDB: 6MBT) and RRAS2-GDP structures (PDB: 2ERY) onto their respective p110α complexes shows RMSDs of 1.15 Å and 0.90 Å, respectively, highlighting conformational changes in the switch regions upon activation and effector binding.

### Structural analysis of KRAS interaction with RAS-binding domain of PI3Kα and CRAF

Amino acid sequence alignment of the RBDs of PI3Kα and CRAF reveals only four conserved residues (Fig. 5D, E). PI3Kα-RBD contains 25 additional residues compared to CRAF-RBD, resulting in elongated rβ1 and rβ2 strands and two extra α-helices (rα2 and rα3). Among the residues interacting with RAS proteins, only K204 and Q205 in PI3Kα are conserved across both effectors. Structural overlay of the KRAS-CRAF and KRAS-p110α shows that the central β-sheet of the two RBDs is rotated by 38 degrees relative to one another[42]. Although the switch-I conformations of KRAS bound to PI3Kα and CRAF are similar, the α2 helix in the switch-II region and the C-terminal end of the α3 helix in the CRAF complex are positioned closer to the RBD compared to their p110α counterparts, underscoring the distinct roles of these regions in effector interactions. Additionally, there are more H-bond and salt-bridge interactions at the KRAS-CRAF RBD interface compared to the KRAS-p110α RBD interface, providing a basis for the stronger binding affinity of KRAS for CRAF compared to PI3Kα.

### Sequence and structural variations in RBD of PI3K isoforms modulate RAS/Rho GTPase binding

To investigate the unique specificity and affinity of the four class I PI3K isoforms towards small GTPases, we analyzed their RBD sequence and structure, revealing limited similarity among them (Fig. 6A). Notably, p110α's RBD stands out with two key features: first, a 4–6 residue insertion between the rβ1 and rβ2-strands crucial for RAS GTPase interactions; second, a distinct 4–7 residue insertion post the rα1-helix, forming a distinctive loop preceding the rα2-helix, implicated in RAS binding in both PI3Kα and PI3Kγ isoforms (Fig. 6B–D). These compositional and conformational differences in the RAS-interacting region across these PI3K isoforms likely contribute to their diverse specificities and affinities towards RAS and Rho GTPases.

### Structural comparison between RRAS2-p110α and HRAS-PI3Kγ complexes

PI3Kα and PI3Kγ exhibit comparable binding affinities to RRAS2 and KRAS, despite sharing only 18% sequence identity in the RBD (Fig. 1D). Among the RAS-interacting residues, only T208, K210, and K227 (PI3Kα numbering) are conserved between these two PI3K isoforms

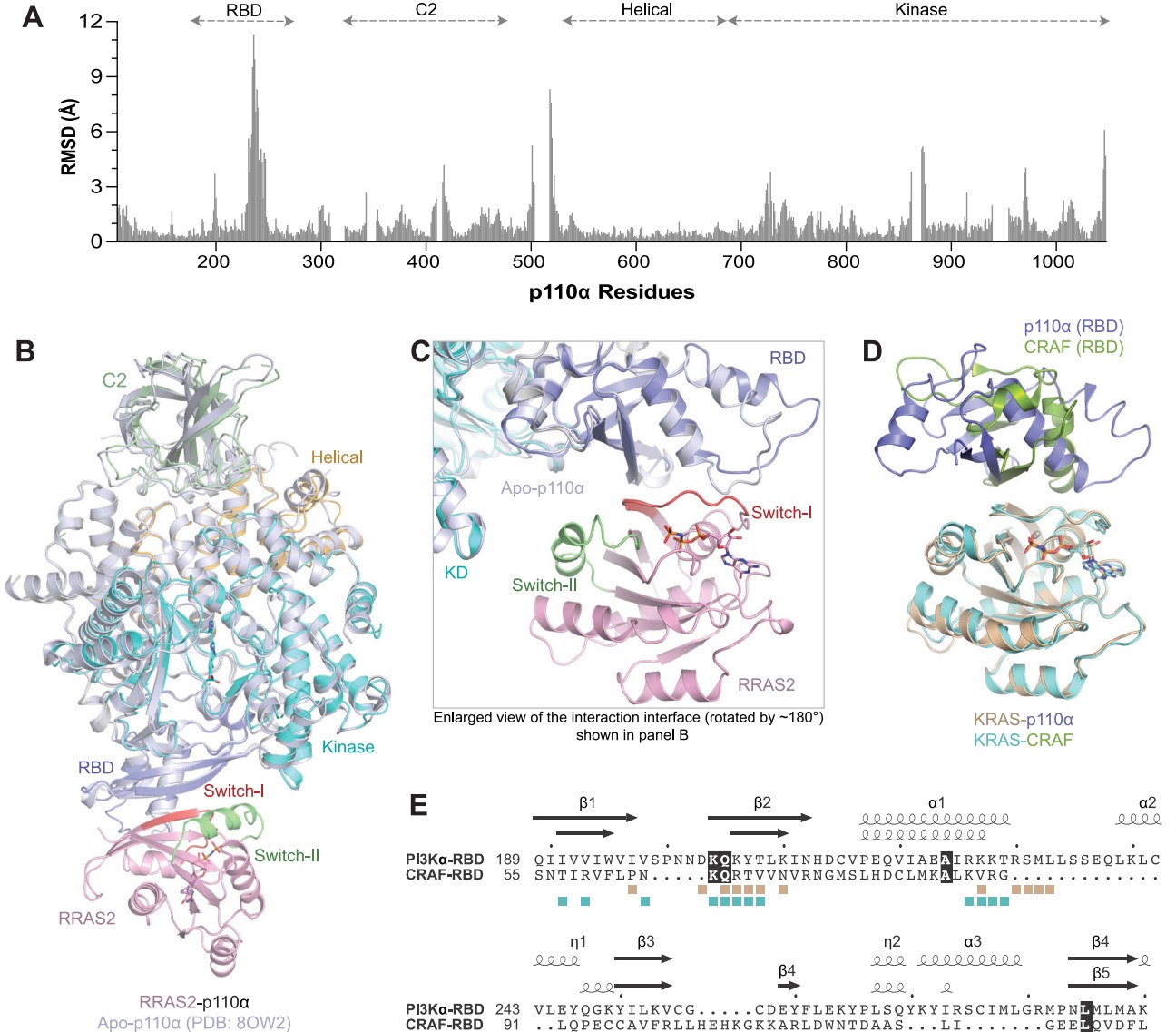

**Fig. 5 | Comparison of apo and RAS-bound structures of p110α, and RAS interactions with p110α and CRAF. A** Bar graph displaying RMSD per residue, comparing apo-p110α (PDB: 8OW2) with p110α complexed with RRAS2. **B** Structural superposition of apo-p110α (PDB: 8OW2) with p110α complexed with RRAS2, aligned using RBD, shows minimal conformational changes in other domains of p110α. **C** Enlarged view of the interaction interface from panel B showing conformational changes in the rα2-helix and rα1-rα2 loop of the p110α-RBD upon binding to RAS. **D** Structural superposition of the KRAS-p110α and KRAS-

CRAF (RBD) (PDB: 6VJJ) complexes, aligned via their respective RAS proteins, reveals structural differences between the CRAF and PI3Kα RBDs, with PI3Kα-RBD exhibiting additional secondary structures due to 25 extra residues. **E** Structure-based sequence alignment of RBDs from CRAF and PI3Kα, with secondary structures annotated above the aligned sequences. Orange and cyan squares mark the KRAS residues interacting with p110α and CRAF, respectively. Fully conserved residues are highlighted in black.

(Fig. 6A and Supplementary Fig. 19A–D). Unique interactions in PI3Kα arise from the insertion between the rβ1-rβ2 strands and the rα2 helix. A notable difference is R230 in PI3Kα (K254 in PI3Kγ), which causes an 18-degree rotational shift in RAS, altering switch-I orientation compared to HRAS-PI3Kγ complex[35] (Fig. 6E–G). The R230A mutation in PI3Kα completely abolishes RAS binding, whereas the K254A mutation in PI3Kγ results in a ten-fold reduction in binding affinity to RAS[35], suggesting divergent roles for these residues in RAS interaction in PI3Kα and PI3Kγ.

In the HRAS-PI3Kγ complex, the nearest atoms between the kinase domain's C-lobe (E919) and switch-II (R73) are ~4.57 Å apart[35], whereas in RAS-p110α complexes, the equivalent residues (E888 in p110α and R73/R83/R84 in KRAS/MRAS/RRAS2) are 9–12 Å apart (Fig. 3E and Supplementary Fig. 8C). In RRAS2-p110α complexes, p110α residues K884, N885, K886, and E888, located in the interhelical turn

between kα5 and kα6 ("k" refers to kinase domain), face RRAS2 residues E80, Q81, and R84 on the α2-helix (switch-II) and R113 on the α3-helix. The side chain density of these residues is not well resolved, suggesting inherent flexibility and a lack of direct interaction between them. The shortest distances observed between these residues in different PI3Kα-RAS complexes vary from 7.5–10 Å. Given the variability in the distance between the kinase domain and the switch-II region of RAS residues across three structures, we cannot rule out potential structural rearrangements that could bring the kinase domain and switch-II region closer when the RAS-PI3Kα complex is positioned on the membrane surface.

Two notable differences explain the lack of direct interaction between the switch-II region and the kinase domain in the RAS-p110α complexes. Firstly, a single residue deletion in p110α between kα5 (residues 877–884) and kα6 (residues 887–895)

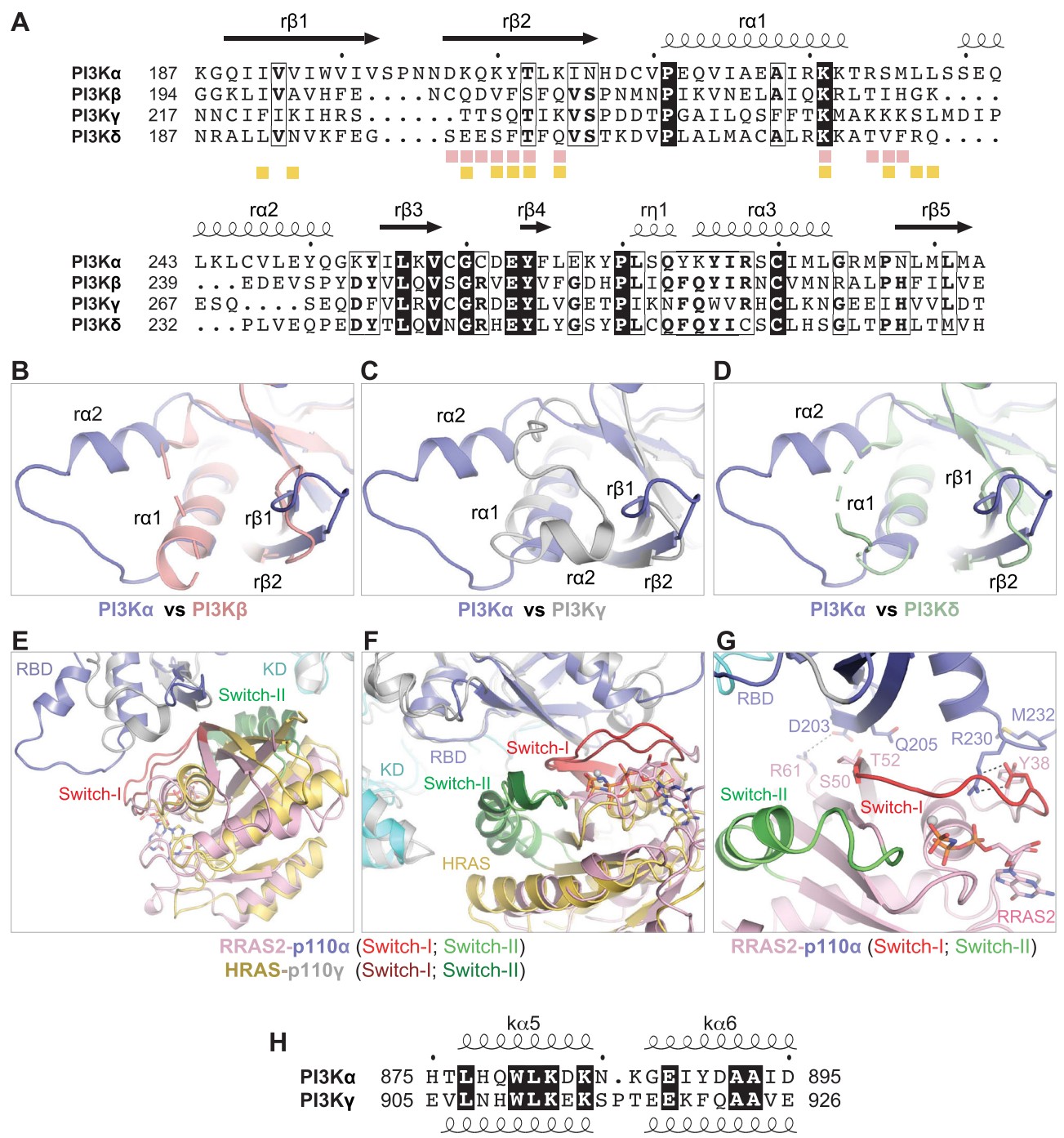

**Fig. 6 | Comparative analysis of RBD in class I PI3K isoforms and RAS complexes with p110α and PI3Kγ. A** Structure-based sequence alignment of the RBD region in four class I PI3K isoforms. Fully conserved residues are highlighted in black, and partially conserved residues are boxed. Pink and yellow squares denote p110α and PI3Kγ (PDB: 1HE8) residues interacting with RRAS2 and HRAS, respectively. **B**−**D** Overlay of the p110α-RBD (blue) with RBD of (**B**) p110β (salmon; PDB: 4BFR), (**C**) PI3Kγ (gray; PDB: 1HE8), and (**D**) p110δ (green; PDB: 7LM2). **E** Overlay of the crystal structure of the RRAS2-p110α complex with the HRAS-PI3Kγ complex (PDB: 1HE8), aligned using RBD from both PI3K isoforms, revealing differences between RAS proteins and the RBD of p110α and PI3Kγ. **F** A 90-degree rotated view from panel E shows variations in the switch regions of RAS proteins and the RBD and kinase domains of PI3Kα and PI3Kγ. The RBD and kinase domains are blue and cyan in PI3Kα and gray in PI3Kγ. **G** Detailed view of the interaction interface in the RRAS2-p110α complex, highlighting interactions unique to the RRAS2-p110α complex as shown in panel A. **H** Amino acid sequence alignment of residues in the kα5 and kα6 helices of PI3Kα and PI3Kγ reveals a deletion in the interhelical turn of PI3Kα.

shortens the interhelical turn, increasing the distance between the kinase domain and the switch-II region (Fig. 6H). Secondly, the inward positioning of the α2 helix in the switch-II region of RAS proteins bound to PI3Kα, compared to HRAS bound to PI3Kγ, further increases this gap (Fig. 6F). Overall, the comparative structural analysis of the RAS-p110α and RAS-p110γ complexes elucidates PI3K isoform-specific interactions involving the switch regions, RBD, and kinase domain.

**Structural comparison of RAS-p110α complex and p110α-p85 complexes with and without the nSH2 domain**

A network of interactions between the catalytic p110α and regulatory p85 subunits maintains PI3Kα in an inactive state[31–34]. Like p110α, p85 comprises five domains: SH3 and BH/GAP domains in the first half, and two SH2 domains (N-terminal nSH2 and C-terminal cSH2) separated by an inter-SH2 (iSH2) domain in the second half (Fig. 2A). The nSH2 and iSH2 domains of p85 form inhibitory contacts with p110α. All p110α-p85 complex structures feature the iSH2 domain; however, the nSH2 domain is disordered in some, resulting in PI3Kα complexes that contain either both nSH2 and iSH2 (niSH2) or solely iSH2. In these structures, the nSH2 domain forms inhibitory contacts with the C2, helical, and kinase domains of p110α, while the iSH2 interacts with the ABD, C2, and activation loop of the kinase domain[29–34]. A regulatory arch within the kinase domain comprising the kα8-kα11 helices (residues 975–1047)[43,44] encircles the catalytic and activation loops, interfacing with the nSH2 and cSH2 domains of p85, and mediates inhibition of p110α's catalytic activity. This regulatory mechanism in class-I PI3K isoforms is distinguished by the absence of the cSH2 inhibitory interaction in the p110α isoform, illustrating isoform-specific regulatory nuances in this PI3K family[45,46]. Full activation of the PI3Kα complex requires binding to GTP-loaded RAS and engagement of the nSH2 and cSH2 domains of p85 by bis-phosphorylated tyrosine (pY) motifs in RTKs[21].

In PI3Kα (p110α + niSH2 p85) structure (PDB: 4OVU), the activation loop interacts with the negatively charged acidic motif of the nSH2 domain through its basic residues (KRER), causing the loop to collapse into a conformation where the initial basic residues form a helical structure, and the C-terminal residues form a U-shaped motif[43] (Fig. 7A, B). Structural comparison of RAS-p110α complexes with p110α-p85(niSH2) complexes showed that the activation loop is disordered in the RAS-p110α complex, likely due to increased flexibility in the absence of inhibitory interactions formed by the nSH2 domain of p85. In PI3Kα structure (p110α + iSH2 p85) without the inhibitory nSH2 domain (PDB: 5DXH), the regulatory arch of the kinase domain undergoes structural changes, with kα11 transitioning from an IN to an OUT conformation, altering local hydrophobic interactions, thereby facilitating the extended activation loop conformation during PI3Kα activation[43] (Fig. 7C, D). Structural comparison of RAS-p110α complexes with p110α-p85(iSH2) shows similar flexibility and disorder in the activation loop in both structures due to the lack of inhibitory interactions by nSH2. Interestingly, in RAS-p110α complexes, the conformation of kα11 in the regulatory arch and U-motif resembles that observed in PI3Kα (p110α + niSH2 p85) structures containing the nSH2 domain. Conversely, the activation loop conformation in the RAS-p110α complex is similar to PI3Kα structures containing only the iSH2 domain, devoid of nSH2 inhibitory interactions.

## Discussion

Since the identification of PI3Kα as a RAS effector three decades ago[47,48], structural elucidation of the RAS–PI3Kα complex has been hindered by the weak binding affinity of classical RAS proteins (H/N/KRAS) to PI3Kα. Utilizing quantitative binding affinity assessments with RAS family GTPases, aided by a high-affinity point mutant or a glue compound, enabled the formation of stable complexes, enabling structural resolution of RRAS2, MRAS, and KRAS with p110α. Structural and mutational analyses of RAS-p110α complexes reveal key interacting residues and compositional differences across KRAS, RRAS2, and MRAS that influence their affinities for p110α. The lack of conservation of key interacting residues in other RAS family GTPases explains their inability to bind to PI3Kα.

Class I PI3Ks are activated by various upstream mechanisms, including RTKs, GPCRs, and RAS GTPases, demonstrating considerable plasticity in their activation[2]. Previous studies have investigated which RAS family GTPases activate the lipid kinase activity in p110α, p110β,

p110δ, and p110γ by measuring PIP3 levels in intact cells[1]. These studies revealed that classical RAS and RRAS subfamily members potently activate p110α and p110γ, with mainly RRAS and RRAS2 capable of activating p110δ, while none of the tested RAS family GTPases could activate p110β. Our binding studies support that only p110α, p110δ, and p110γ are activated by RAS family GTPases[1].

The RAS-p110α complex structures described herein show that RAS interaction with p110α occurs primarily via RBD, with no direct interaction with the C-lobe of the kinase domain, unlike the HRAS-PI3Kγ complex[35]. However, potential interactions between switch-II and the kinase domain's C-lobe when the RAS-PI3Kα complex is membrane-bound cannot be ruled out. Our findings corroborate previous single-molecule and biochemical studies suggesting that RAS-mediated PI3Kα activation is primarily driven by increased membrane recruitment[29,49]. These observations also align with the activation mechanism of the H1047R mutation, where the mutated residue aids in positioning the kinase domain's C-lobe near the membrane, enabling PI3Kα activation in an RAS-independent manner.

An open question in the field is which RAS family GTPases activate PI3Kα during normal and oncogenic signaling, likely influenced by factors like affinity, cellular distribution, expression levels, and population of active RAS GTPases. Despite significant conservation in the G-domain, classical RAS and RRAS subfamily GTPases show variations in the effector binding motif and hypervariable region (HVR), affecting their effector affinity and localization. RRAS2's oncogenic role, linked to its activation of the PI3K-AKT pathway[16], is underscored by the significant enhancement of PI3K-dependent signaling following ectopic expression of RRAS2-G23V[19]. RRAS2 is crucial in cancer cells with constitutively active classical RAS proteins, indicating these GTPases are not functionally redundant. In KRAS-transformed breast cancer cells, RRAS2 activates a subset of PI3Kα, promoting PI3K-dependent tumorigenesis[16]. It is suggested that RRAS2 and classical RAS proteins function independently, with RRAS2 driving the PI3Kα/AKT axis and classical RAS proteins activating the RAF/MEK/ERK pathway[19]. RRAS subfamily GTPases may activate PI3Kα during normal signaling due to their higher affinity than classical RAS proteins. However, in oncogenic signaling involving mutated classical H/N/KRAS proteins, a higher concentration of active GTP-bound proteins may compensate for their reduced affinity, potentially activating PI3Kα.

The interaction between p110α and RAS and the p85-mediated release of autoinhibition enables PI3Kα activation. To gain structural insights into RAS-mediated activation of PI3Kα at the membrane, we generated a structural model of the fully active PI3Kα complex (p110α and p85) bound to active RAS and phosphorylated RTKs, incorporating mechanistic insights from previously reported models[21,29,34,43,44]. This model integrates structural data from the RAS-p110α complex described in this study with previously reported structures of p110α in complex with p85 (iSH2) (PDB: 4OVV), and the nSH2 and cSH2 domains of p85 bound to phospho-tyrosine-containing peptides[50,51] (PDB: 2IUI and 5AUL) (Fig. 8A). This structural representation elucidates the molecular interactions and structural changes governing PI3Kα activation, emphasizing p85's role in relieving autoinhibition by binding phospho-tyrosines, and the involvement of RAS proteins in positioning the kinase domain at the membrane through interaction with the p110α-RBD. Upon binding an RTK-phosphopeptide, the nSH2 domain disengages from p110α, breaking its inhibitory contact with the kinase domain's regulatory motif and disrupting its contact with the helical domain[30]. After nSH2 release, the kinase domain's C-lobe moves away from the C2 domain, exposing the p110α membrane-binding surface. Our structural data suggest that the primary role of the membrane-anchored RAS proteins is to position the p110α kinase domain at the membrane surface and potentially establish long-range interactions with the kinase domain's C-lobe to facilitate PIP2 substrate access to the active site for enzymatic reaction. Membrane engagement of p110α causes the ABD to detach from the catalytic core and the C2

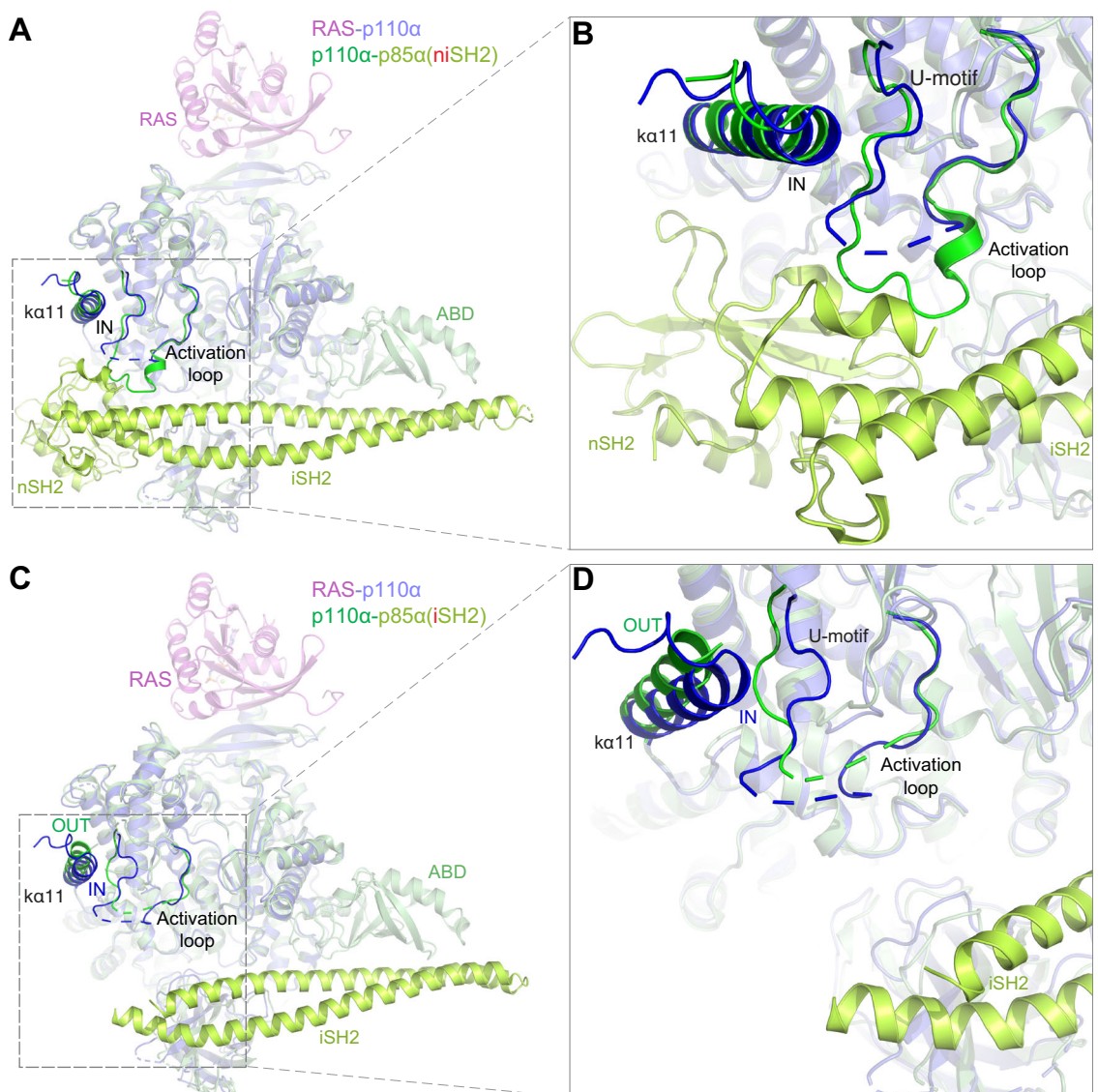

**Fig. 7 | Comparative structural analysis of the RAS-p110α complex versus the p110α-p85 complexes, with and without the inhibitory nSH2 domain.**
**A** Structural superposition of p110α in the RAS-p110α complex and the p110α-p85 (niSH2) complex (PDB: 4OVU) containing nSH2 domains reveals conformational differences in the activation loop and similarities in kα11 and the U-motif.
**B** Enlarged view of kα11, U-motif, and the activation loop with the inhibitory nSH2 domain as shown in panel A. **C** Structural superposition of p110α in the RAS-p110α complex and the p110α-p85 (iSH2) complex (PDB: 5DXH) without the inhibitory nSH2 domain reveals conformational similarities in the activation loop and differences in kα11 and the U-motif. **D** Enlarged view of kα11, U-motif, and the activation loop without the inhibitory nSH2 domain as shown in panel C.

domain to separate from the iSH2 domain of p85[30,31]. Recent Cryo-EM analysis revealed complete dissociation of the ABD and regulatory subunit from the catalytic core upon phosphopeptide binding[31], with HDX-MS results showing similar observation only upon membrane binding[30]. Release of autoinhibition and membrane binding increases the flexibility of the kinase domain's activation loop and positions it near ATP, generating an active PI3Kα conformation for substrate catalysis. This orientation facilitates PIP2 binding to the kinase domain's substrate binding site, catalyzing its conversion to PIP3.

Over 80% of mutations in *PIK3CA* occur at three hotspots[2]. The E542K and E545K mutations are located in the helical domain, whereas the H1047R mutation is in the kinase domain[52]. Previous studies have demonstrated that the H1047R mutant does not rely on RAS proteins for activation but depends on the release of p85-mediated autoinhibition[53]. Conversely, E542K and E545K mutants need RAS for activation and do not require binding the phospho-

tyrosine motif of RTKs with the nSH2 and cSH2 domains of p85 to release the autoinhibition[53]. Structural data show that E542K and E545K mutations disrupt salt-bridge interactions between the nSH2 and helical domains[32,44], resulting in the loss of most p110α-p85 interactions, except for the iSH2 domain remaining bound to the ABD. The ABD exhibits enhanced mobility in these structures, presumably due to a hinge mechanism linking it to the rest of p110α. These helical domain mutations mimic enzyme activation by a phospho-peptide, and the oncogenic transformation driven by helical domain mutations is unaffected by the deletion of the ABD−p85 complex[54]. The oncogenic activity of E542K and E545K in cell culture is RAS-dependent, and an RBD-inactivating mutation nullifies their transforming ability[31,54]. The schematic in Fig. 8B depicts the mechanism of activation of helical mutations, where the mutated residue releases the autoinhibition by p85 and relies solely on RAS interaction for PI3Kα activation.

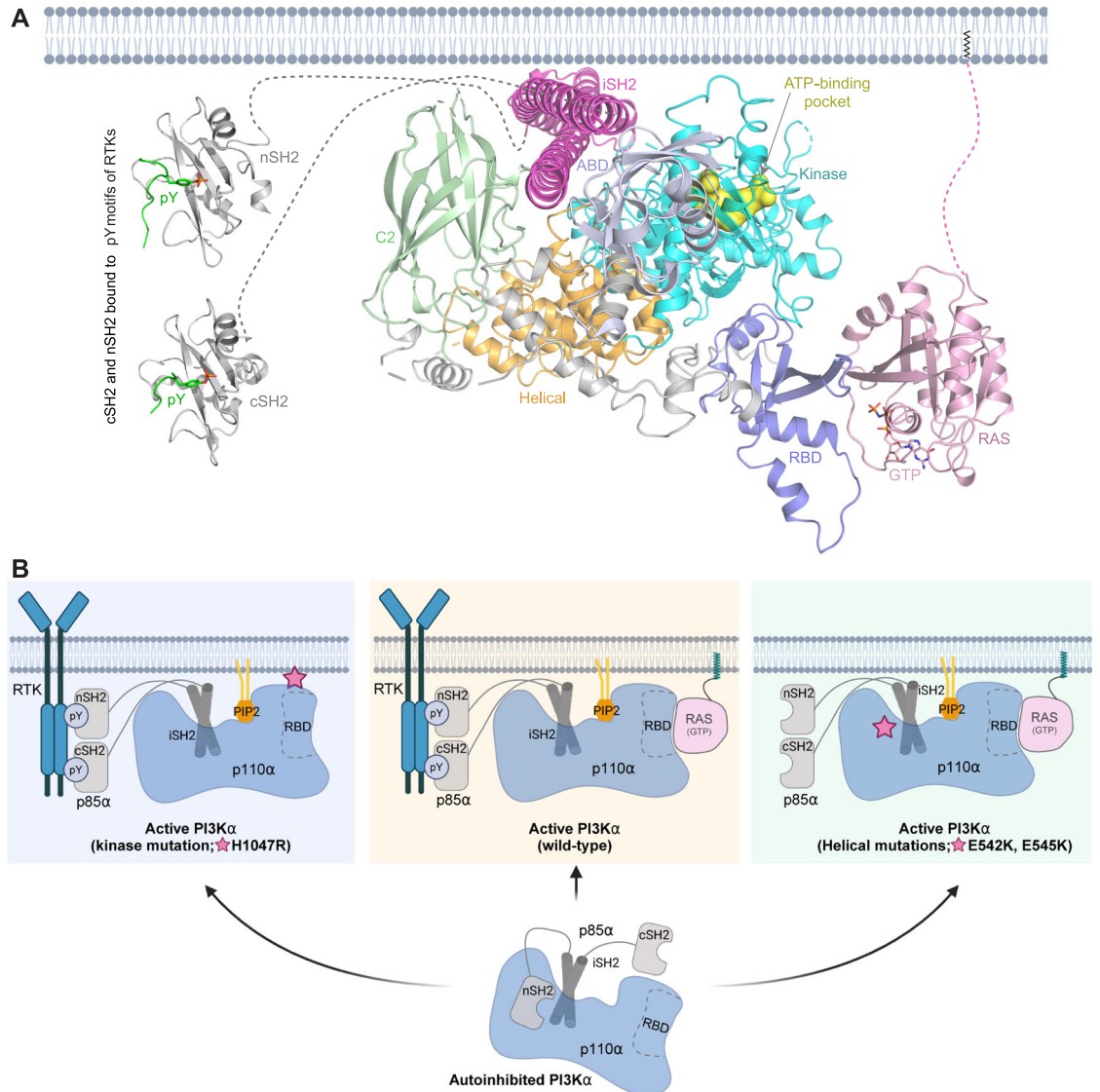

**Fig. 8 | Model of fully active RAS-PI3Kα complex on the plasma membrane and schematic of PI3Kα activation in normal and oncogenic conditions. A** Model of RAS bound to PI3Kα, generated by aligning the RRAS2-p110α complex on the apo-structure of the p110α-p85(iSH2) complex (PDB: 4OVV) and incorporating the structures of nSH2 and cSH2 domains of p85 solved in complex with the pY motifs of receptor tyrosine kinases (PDB: 2IUI and 5AUL). The binding of the nSH2 and cSH2 domains to the pY motifs of RTK relieves the inhibitory effect of p85 on p110α. The membrane-anchored RAS (pink), upon binding to p110α-RBD, positions PI3Kα at the membrane, thereby facilitating access of membrane-bound PIP2 to the active site for the catalytic reaction. **B** Schematic illustration depicting the activation mechanism of wild-type PI3Kα and its variants with helical and kinase domain mutations. Created in BioRender. Simanshu, D. (2024) https://BioRender.com/v62l361.

Structural and HDX-MS analysis of the kinase domain mutant H1047R show alterations in the membrane-binding surface and activation loop interactions[30,31,33]. The H1047R mutation, situated at the end of the kα11-helix, causes a ~90-degree rotation, orienting the C-terminus region towards the membrane surface[33]. In PI3Kα-H1047R structures, the region (residues 1050–1062) C-terminal of H1047R is unstructured, indicating heightened flexibility and membrane inter-action via the lipid-binding WIF motif[30,33]. Therefore, H1047R's functional gain stems from increased membrane binding and a catalytically active activation loop conformation. Fig. 8B depicts how H1047R mutation enables PI3Kα activation independently of RAS, mimicking the role carried out by RAS proteins.

The clinical development of PI3K inhibitors has historically been hindered by on-target toxicity[4]. The therapeutic efficacy of alpelisib, a US FDA-approved orthosteric PI3Kα inhibitor used in combination with fulvestrant, is limited by a narrow therapeutic index and various

resistance mechanisms[55]. Given the limited sequence homology across the RBD of class I PI3K isoforms, targeting the isoform-specific glue binding site within PI3Kα-RBD offers a promising strategy to selectively inhibit the PI3Kα-RAS interaction. Structural analysis of the KRAS-p110α complex revealed that the glue compound D927 binds to an induced pocket in the p110α RBD and interacts with KRAS residues R41 and Y40, leading to the hypothesis that modifying the RAS-interacting group of D927 to create a steric clash with Y40 could disrupt the KRAS-p110α interaction. Extensive structure-based drug design and molecular modeling efforts have led to the development of a RAS-PI3Kα molecular breaker compound (BBO-10203), which binds near the RAS-p110α RBD interface, effectively blocking interaction with classical RAS proteins[56]. This selective blockade preserves the essential kinase function of PI3Kα, minimizing adverse events like hyperglycemia associated with kinase activity. The effects of BBO-10203 on cancer cell signaling inhibition and transcriptional regulation

closely resemble those of alpelisib. The development of the RAS-PI3Kα breaker provides a foundation for creating similar inhibitors that target the RBDs of other effector proteins, thereby preventing their activation by RAS-like GTPases.

## Methods

### DNA for protein production

Expression clones for wild-type and point mutants of *HRAS*(1-169) (UniProt: P01112), *KRAS*(1-169) (UniProt: P01116), *NRAS*(1-169) (UniProt: P01111), *RRAS*(22-192) (UniProt: P10301), *RRAS2*(1-179), *RRAS2*(11-179) (UniProt: P62070), *MRAS*(1-178), *MRAS*(10-178) (UniProt: O14807), *RIT1* (2-219) (UniProt: Q92963), and *PIK3CA*-RBD(157-289) (UniProt: P42336) from human were synthesized (ATUM, Inc.) for creating Gateway Entry clones optimized for *E. coli* expression. These Entry clones were then subcloned into pDest-566 (Addgene #11517) to produce final expression clones containing the target protein with an amino-terminal His6-MBP (maltose-binding protein) fusion. For producing human *PIK3CA*(105-1068) W1057A/I1058A/F1059A (UniProt: P42336) in insect cells, a Gateway Entry clone was synthesized (ATUM, Inc.). The Entry clone was transferred to a baculovirus expression vector containing amino-terminal His6-MBP fusions via Gateway LR recombination (Thermo Fisher Scientific) into pDest-636 (baculovirus, Addgene #159574). An expression clone for human *PIK3CG*(144-1102)-His6 (UniProt: P48736) was generated using the procedure as detailed by Whitely et al.[57] for *PIK3CG*(144–1102)V223K-His6. Final baculovirus expression clones were used to produce bacmid DNA in the DE95 strain using the Bac-to-Bac system (Thermo Fisher Scientific).

Expression clones for human *PIK3CA*(2-1068)W1057A/I1058A/F1059A + *PIK3R1*(322-600) (UniProt: P42336 and P27986), *PIK3CB*(1-1070)W1060A/M1061A + *PIK3R1*(322-600) (UniProt: P42338 and P27986), and *PIK3CD*(1-1044) + *PIK3R1* (322-600) (UniProt: O00329 and P27986) were created via Gateway multisite reaction in pDest-623 (Addgene #161878) following the methods described by Wall et al.[58]. Specifically, the *PIK3CA* (p110α) expression element was derived from two entry clones: one with a polyhedrin promoter upstream of His6-MBP (Addgene #162933) and the second containing PCR-generated TEV-*PIK3CA*. The *PIK3R1*(p85) elements for all three constructs originated from an entry clone with a polyhedrin promoter upstream of an untagged p85 (optimized for insect cell expression by ATUM, Inc.). The final baculovirus Expression Clone was utilized to generate bacmid DNA using the Bac-to-Bac system (ThermoFisher Scientific).

### Protein expression and purification

All RAS GTPases (except for NRAS) and p110α RBD were expressed in *E. coli* using the previously described Dynamite media protocol[59]. This medium, a modification of Studier's auto-induction media, permits higher cell density after overnight induction with IPTG at 16 °C. NRAS was expressed in *Vibrio natriegens* as described by Nelson et al.[60], while all other PI3K constructs were expressed in insect cells using the methods described by Snead et al.[61]. The RAS GTPases and p110α RBD were purified according to the protocol of Kopra et al.[62], except that MgCl$_2$ was omitted during the p110α RBD purification. Generally, the purification process involved an IMAC (immobilized metal ion affinity chromatography) step to capture the His-tagged protein from the bacterial lysate, followed by proteolytic cleavage of the His6-MBP N-terminal fusion with His6-TEV protease (purified in-house). The target protein was then isolated by a second IMAC step, with the untagged target protein in the column flow-through, while most other contaminants remained His6-tagged and were retained on the column. The final purification step involved size-exclusion chromatography.

PI3Kγ(144-1102)-His6 was purified as described by Kopra et al.[62] for KRAS4b(1-169), with specific modifications: the buffer throughout was 20 mM Tris-HCl, pH 8.0, without MgCl$_2$, and the elution from the second IMAC step employed a 10-column volume (CV) gradient to account for the higher affinity of His-TEV protease compared to the

His-tagged protein. A similar purification strategy was used for these PI3K proteins: p110α(105-1068)W1057A/I1058A/F1059A, p110α(2-1068)W1057A/I1058A/F1059A + p85(322-600), p110β(1-1070)W1060A/M1061A + p85(322-600), and p110δ(1-1044) + p85(322-600) as described in Messing et al.[63]. To purify these proteins, frozen pellets were thawed and resuspended in 20 mM Tris-HCl, pH 8.0, 300 mM NaCl, 1 mM TCEP (Buffer 1), and a 1:200 (v/v) protease inhibitor cocktail (P8849, Sigma Aldrich). For the p110α(2-1068)W1057A/I1058A/F1059A + p85(322-600) constructs containing the ABD and co-expressed with p85, pellets were resuspended in 20 mM Tris-HCl, pH 8.0, 150 mM NaCl, 1 mM TCEP, 5% glycerol, 1% Triton X-100, 1 mM orthovanadate, and a 1:200 (v/v) protease inhibitor cocktail (Buffer 2). Resuspension volumes were 100 mL of buffer per liter of harvested culture. Cells were lysed with two passes through a Microfluidizer M-110EH (Microfluidics Corp.) at 7000 psi. The lysates were clarified by centrifugation at 70,000 x g for 35 min at 4 °C. The clarified supernatants were immediately filtered using a 250 mL Autofil 0.45 μM High Flow PES Bottle Top Filter (Thomas Scientific).

Chromatography was conducted at room temperature (~22 °C) using NGC medium-pressure chromatography systems (BioRad Laboratories Inc.). The lysates were adjusted to 25 mM imidazole and loaded onto equilibrated IMAC columns containing Ni-Sepharose High-Performance nickel-charged resin (Cytiva) at a ratio of 20 mL of resin per liter of culture. Column equilibration buffer was Buffer 1 plus 25 mM imidazole or Buffer 2 in the case of the p110α/p85; imidazole was added to reduce non-specific binding to the purification resin. The columns were washed to baseline with Buffer 1 or 2 plus imidazole. A 10 CV gradient to 500 mM imidazole was implemented, followed by a 2 CV wash with 500 mM imidazole. SDS-PAGE and Coomassie staining were used for elution fraction analysis. Appropriate fractions were pooled, His6-TEV protease at 4 mg/mL was added at a 1:20 (v/v), and the digestion pool was dialyzed using a 10 kDa MWCO membrane (SnakeSkin™ Dialysis Tubing, Thermo Fisher Scientific) against the appropriate buffer without imidazole overnight at 4 °C. Another IMAC was performed using Ni-Sepharose Fast Flow nickel-charged resin (Cytiva) with a similar format (20 mL of resin per liter of culture) but with no imidazole in the equilibration or wash buffers. A 5 CV gradient to 100 mM imidazole was performed. Appropriate fractions were pooled after SDS-PAGE and Coomassie staining analysis and dialyzed using a 10 kDa MWCO membrane against 20 mM Tris-HCl, pH 8.5, 75 mM NaCl, 1 mM TCEP (Buffer 3) for p110α overnight at 4 °C. For the p110α/p85 constructs, the buffer was 20 mM Tris, pH 8.5, and 1 mM TCEP (Buffer 4). The pools were loaded onto an equilibrated HiTrap Q High-Performance anion exchange column (Cytiva) (10 mL resin per liter of culture). Column equilibration was accomplished using 2 CV of either Buffer 3 or 4. Columns were washed with 3 CV of the respective buffers before elution with a 10 CV gradient from the equilibration buffer to 250 mM NaCl, followed by a 2 CV wash with 500 mM NaCl. Appropriate fractions were pooled after SDS-PAGE and Coomassie staining analysis and concentrated using 30 kDa MWCO centrifugation units (EMD Millipore) to an appropriate volume for size exclusion chromatography. The concentrated pool was loaded onto an equilibrated Superdex 200 resin column (Cytiva) in our final buffer of 20 mM Tris-HCl, pH 8.0, 150 mM NaCl, 1 mM TCEP (Buffer 5). Appropriate fractions were pooled using SDS-PAGE and Coomassie staining analysis, concentrated as needed by 30 kDa MWCO centrifugation units, and filtered with a 0.22 μM syringe filter (Millex-GP, EMD Millipore). Protein concentration was determined by measuring A280 (Nanodrop 2000C Spectrophotometer, Thermo Fisher Scientific), and purified proteins were flash-frozen in 250 μL aliquots in 1.5 mL Eppendorf tubes using liquid nitrogen.

### Preparing GMPPNP-bound RAS GTPases by nucleotide exchange

Recombinant RAS GTPases purified from *E. coli* are typically GDP-bound; for structural and binding studies, we performed nucleotide

exchange to replace GDP with slowly hydrolyzable GTP-analog, GMPPNP. RAS-GDP was incubated with a five-fold molar excess of GMPPNP (Jena Biosciences) in a mixture containing 200 mM ammonium sulfate and 100 μM $ZnCl_2$ at protein concentrations of 0.1 to 0.5 mM. The protein stock was diluted to reduce the $MgCl_2$ concentration to below 1 mM in the reaction. Alkaline phosphatase-agarose beads (Sigma) were added at a ratio of 1 unit per milligram of protein and gently rotated at room temperature for 3 h. The beads were then separated by centrifugation at 16,900 x g for 2 min. An additional ten-fold molar excess of GMPPNP and 25 mM $MgCl_2$ were added to the supernatant, which was incubated overnight at 4 °C. Excess GMPPNP was removed using a PD-10 desalting column packed with Sephadex G-25 resin (Cytiva), equilibrated with 20 mM HEPES, pH 7.4, 150 mM NaCl, 5 mM $MgCl_2$, and 1 mM TCEP. The nucleotide exchange efficiency for the RAS GTPases was assessed using high-performance liquid chromatography (HPLC). RAS samples were diluted in 0.1 M $K_2HPO_4$ and 1 mM tetrabutylammonium hydrogen sulfate (buffer A) and injected onto an Ultrasphere 5 ODS, 250 × 4.6 mm column (Hichrom) on a Waters e2695 Alliance HPLC System (Waters). Bound nucleotides were eluted with a linear gradient of 30% acetonitrile in buffer A at a 0.6 mL/min flow rate. GDP and GMPPNP standards were run to confirm the identities of the nucleotides bound to the RAS proteins, which exhibited almost 100% GMPPNP loading after the exchange.

## Synthesis of the molecular glue compound D927

The compound D927 (2-[3-Fluoro-4-({7-[2-(2-methoxyethoxy)phenyl] thieno[2,3-d]pyridazin-4-yl}amino)phenyl]acetamide) was synthesized following the methods previously outlined in Tsuji et al. for compound 26b[39] (see Scheme 1 in Supplementary Information). Briefly, D927 was synthesized in three steps, starting from commercially available 4,7-dichlorothieno[2,3-d]pyridazine. In the first step, an SNAr reaction with 4-amino-3-fluorophenylacetic acid yielded a mixture of regioisomers, which were separated by C18 reverse-phase preparative HPLC to obtain compound 2 with a 32% yield. The 2-(2-methoxyethoxy) phenyl group was then introduced via Suzuki coupling using Pd(PPh3)4, resulting in compound 3 with a 71% yield. Finally, the ethyl ester was replaced by ammonia under microwave irradiation to yield compound 4 (D927) with a 34% yield (overall yield: 7.7%).

## Complex formation, crystallization, and data collection

For crystallization of the RRAS2-p110α complex, p110α(105-1068) at a concentration of 12 mg/mL was combined with GMPPNP-bound RRAS2(1-179)-Q36A at 2.5 mg/mL in a 1:1.5 ratio. The PI3Kα kinase region was stabilized by adding a 2-molar equivalent of the PI3Kα-selective inhibitor GDC-0326, followed by a 30 min incubation at room temperature. Crystallization screening was performed at 20 °C using the sitting-drop vapor diffusion method, mixing the RRAS2-PI3Kα mix with an equal volume of reservoir solution (200 nL:200 nL). Crystals emerged within 72 h under Morpheus II (Molecular Dimensions) condition F11, utilizing a buffer system of 0.1 M Gly-Gly/AMPD at pH 8.5, 0.1 M monosaccharides II mix, and 50% v/v of Morpheus precipitant mix 7 containing 20% w/v PEG 8000, 40% v/v 1,5-Pentanediol. Optimization included adjusting the pH (0.1 M Gly-Gly/AMPD, pH 8.3) and reducing the precipitant mix concentration to 48% v/v, resulting in crystals diffracting to 3.5 Å. Matrix microseeding was employed to further enhance diffraction quality, where crystals were transferred to a seed bead tube (Hampton Research), vortexed for 30 s, and subjected to crystallization screening with a drop ratio of 200 nL protein mix, 133 nL well solution, and 67 nL of seeds. This led to the identification of 10 more conditions, with one yielding crystal diffracting to 3.1 Å, under Morpheus screen condition A10, containing 0.1 M (Bicine/Tris) at pH 8.5, 0.06 M divalent mix, and 30% Morpheus precipitant mix 1 (40% v/v PEG MME 500, 20% w/v PEG 20,000).

For the MRAS-p110α complex, similar procedures were followed, with purified p110α(105-1068) at 12 mg/mL combined with GMPPNP-bound MRAS(1-178)-Q35A at 2.5 mg/mL at a 1:1.5 ratio. To stabilize the p110α kinase region, a 2-molar equivalent of GDC-0326 was added to the protein mix, followed by a 30-min incubation at room temperature. Sparse matrix screening was employed, resulting in crystals appearing within 72 h under Morpheus III condition G10, utilizing a buffer system of 0.1 M Tris/Bicine at pH 8.5, 1.2% w/v cholic acid mix, and 50% v/v Morpheus precipitant mix 2 containing 40% v/v ethylene glycol and 20% w/v PEG 8000. The crystal diffracted to a resolution of 2.75 Å.

For crystallization and structure determination, KRAS-p110α complexes were formed by combining purified p110α(105-1068) with GMPPNP-bound KRAS(1-169) at a 1:1.1 stoichiometric ratio, reaching final concentrations of 10 mg/ml and 1.8 mg/ml, respectively. Glue compound D927, dissolved in 100% DMSO, was added to the mixture at a 1.3-molar equivalent. After a 30 min incubation at room temperature, crystallization screening was performed at 20 °C using the sitting-drop vapor diffusion method. Crystals of KRAS-p110α complex with the glue compound D927 appeared within 72 h under conditions of 0.1 M Tris at pH 8.0, 0.1 M NaCl, and 10% w/v PEG 20,000, diffracting to a resolution of 2.81 Å.

Furthermore, to crystallize active RRAS2, purified GMPPNP-bound RRAS2(11-179) at a concentration of 16.5 mg/mL was utilized in a sparse matrix screen, resulting in crystal formation within 12 h under PGA screen condition G2, containing 0.1 M Tris, pH 7.8, 5% w/v PGA-LM, and 30% w/v PEG MME 550, with the crystal diffracting to 1.45 Å. Similarly, for active MRAS crystallization, purified 18 mg/mL GMPPNP-bound MRAS(10-178) was subjected to a sparse matrix screen, leading to crystal appearance within 12 h under Morpheus II screen condition B5, featuring a buffer system of 0.002 M Divalents II mix and 50% v/v Morpheus precipitant mix 5 containing 30% w/v PEG 3000, 40% v/v 1,2,4-butanetriol, and 2% w/v NDSB 256. The crystal diffracted to 2.10 Å. No additional cryoprotectant was added to the crystals before freezing because the conditions from which the crystals were derived contained cryoprotectants. Diffraction data was collected on beamlines 24-ID-C/E at the Advanced Photon Source (Argonne) and beamline 14-1 at the Stanford Synchrotron Radiation Light source (SLAC).

## Structure determination

Crystallographic datasets were processed using XDS[64], and Matthew's coefficient was determined using Xtriage[65] (Phenix v1.21). The structures of RRAS2-p110α and MRAS-p110α complexes were solved by molecular replacement using the program Phaser as implemented in the Phenix suite of programs, using GDP-bound RRAS2 (PDB: 2ERY), GMPPNP-bound MRAS (PDB: 7TVF) and human p110α (PDB: 5DXT) as search models[66]. Similarly, the structure of the KRAS-p110α complex was determined using GMPPNP-bound KRAS (PDB: 6VC8) and human p110α (PDB: 5DXT) as search models by molecular replacement using the program Phaser (Phenix v1.21). GMPPNP-bound RRAS2 and MRAS structures were determined using the GMPPNP-bound MRAS and RRAS2 structures present in their respective complexes with p110α as a search model. The initial solution obtained from molecular replacement was refined using the program Phenix.refine[65] (Phenix v1.21), and the resulting Fo-Fc map showed apparent electron densities for the GMPPNP nucleotide, RRAS2/MRAS, and p110α domains. The model was further improved using iterative cycles of manual model building in Coot[67] (Coot v0.9.8.8) and refinement with Phenix.refine[65]. The GMPPNP nucleotide was placed in the nucleotide-binding pocket of RAS proteins, followed by the placement of glue compounds in the KRAS-p110α complex. Iterative build omit $\sigma_A$-weighted 2mFo-Fc electron density map was generated using Phenix.Autobuild[68]. The solvent molecules were added by the automatic water-picking algorithm in Coot. These solvent molecules were manually checked during model building until the final round of refinement was completed. Data collection and refinement statistics are shown in Table 1.

## Structural and sequence analysis

Secondary structural elements were assigned using DSSP (https://swift.cmbi.umcn.nl/gv/dssp/). Amino acid sequence alignments were performed utilizing Clustal Omega[69]. Pairwise structural alignments were performed utilizing the pairwise structural alignment tool (rcsb.org/alignment). Figures depicting sequence alignments were generated using ESPript[70]. Protein-protein interaction interfaces were illustrated schematically using PDBsum[71]. Figures were generated by PyMol (Schrödinger, LLC). The degree of rotation observed between protein complexes was determined by the RotationAxis script for PyMol (https://pymolwiki.org/index.php/RotationAxis). The SBGrid Consortium provided support for crystallographic and structural analysis software[72]. Phylogenetic analysis was performed using the amino acid sequences of the RAS GTPase family of proteins. A maximum likelihood tree was constructed using MEGA11 with a Le/Gascuel (LG) substitution model (with a discrete gamma distribution of 5 categories) and 500 bootstrap replicates[73,74].

## Binding affinity measurement using isothermal titration calorimetry (ITC) assay

Proteins for the ITC assay were prepared by dialyzing overnight at 4 °C against a buffer containing 20 mM HEPES (pH 7.4), 150 mM NaCl, 5 mM $MgCl_2$, and 1 mM TCEP (dialysis buffer). Duplicate ITC measurements were conducted using a MicroCal PEAQ-ITC instrument (Malvern Panalytical) at 25 °C. Each ITC experiment involved 30–50 μM of either wild-type or mutant PI3K isoforms in the cell, stirred at 550 rpm for PI3Kα, β, γ, and δ, or 750 rpm for PI3Kα-RBD. We used 300–500 μM of wild-type or mutant RAS proteins in the syringe, delivered across 19 injections of 2 μL each at 175 s intervals. For ITC experiments utilizing glue compound (D927), a 500 μM stock solution of the glue compound was prepared in the dialysis buffer containing 5% DMSO. 1000 μM KRAS and 100 μM PI3Kα-RBD were diluted by half with the stock solution of the glue compound. ITC experiment involved 50 μM PI3Kα-RBD with 250 μM glue compound in the dialysis buffer containing 2.5% DMSO in the cell, and 500 μM KRAS with 250 μM glue compound in the dialysis buffer containing 2.5% DMSO in the syringe. Data analysis was conducted using a binding model incorporating the "one set of sites" approach, employing a nonlinear least-squares algorithm integrated into the MicroCal PEAQ-ITC analysis software (v1.41, Malvern Panalytical). All binding data are tabulated in Supplementary Tables 1, 2, and 3.

## Reporting summary

Further information on research design is available in the Nature Portfolio Reporting Summary linked to this article.

# Data availability

Data supporting the findings of this manuscript are available from the authors upon request. The atomic coordinates and structure factors have been deposited into the Protein Data Bank and are available under accession numbers 9B4S (RRAS2-p110α complex), 9B4T (MRAS-p110α complex), 9C15 (KRAS-p110α complex with glue D927), 9B4Q (RRAS2 bound to GMPPNP), and 9B4R (MRAS bound to GMPPNP). Structures used as initial models for molecular replacement are available in the PDB under accession codes 2ERY, 7TVF, 5DXT, and 6VC8. Structures used for superpositions and analysis are available in the PDB under accession codes 8OW2, 6GOD, 6MBT, 2ERY, 4OVU, 5DXH, 4OVV, 2IUI, 5AUL, 6VJJ, 1HE8, 4BFR, and 7LM2.

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

## Acknowledgements

We acknowledge Hannah Ambrose, Allison Champagne, Julia Cregger, Matt Drew, Peter Frank, Natalie Granato-Guerrero, Brianna Higgins, Min Hong, Vi Le, Skylar Mackay, Jennifer Mehalko, Ashley Mitchell, Shelley Perkins, Ivy Poon, Nitya Ramakrishnan, Katie Reeley, Amanda Seabolt, Mukul Sherekar, Matt Smith, Kelly Snead, Troy Taylor, Vanessa Wall, and Nicholas Wright for their help with production of recombinant proteins used in this work. We thank Timothy Waybright for performing the nucleotide exchange analysis. X-ray diffraction data were collected at the Northeastern Collaborative Access Team beamlines, which are funded by the National Institute of General Medical Sciences (NIGMS) from the NIH (P30 GM124165). The Eiger 16 M detector on 24-ID-E is funded by an NIH-ORIP HEI grant (S10OD021527). This research used resources of the Advanced Photon Source, a U.S. Department of Energy (DOE) Office of Science User Facility operated for the DOE Office of Science by Argonne National Laboratory under Contract No. DE-AC02-06CH11357. Additionally, this research used resources of the Stanford Synchrotron Radiation Lightsource, SLAC National Accelerator Laboratory, which is supported by the U.S. Department of Energy, Office of Science, Office of Basic Energy Sciences under Contract No. DE-AC02-76SF00515. The SSRL Structural Molecular Biology Program is supported by the DOE Office of Biological and Environmental Research and the NIH, NIGMS (P30GM133894). This project was funded in part with federal funds from the National Cancer Institute, National Institutes of Health (NIH) Contract 75N91019D00024. The content of this publication does not necessarily reflect the views or policies of the Department of Health and Human Services, and the mention of trade names, commercial products, or organizations does not imply endorsement by the US Government.

## Author contributions

D.J.C., W.Y., and D.K.S. carried out structural and biophysical studies. S.M., W.G., and D.E. assisted with the recombinant proteins. T.T., M.Y., and S.F. provided the glue compound used in this study. D.M.T., D.V.N., and F.M. contributed to the analysis. DKS coordinated and supervised this project. D.J.C. and D.K.S. wrote the manuscript with input from all co-authors.

## Funding

## Competing interests

All authors, except F.M., have no competing interests. F.M. is a consultant for Ideaya Biosciences, Kura Oncology, Leidos Biomedical Research, Pfizer, Daiichi Sankyo, Amgen, PMV Pharma, OPNA-IO, and Quanta Therapeutics, has received research grants from Boehringer-Ingelheim, and is a consultant for and cofounder of BridgeBio Pharma.
