## [Transparent Peer Review file · Nature Communications]

Structural Insights into Isoform-Specific RAS-PI3K α Interactions and the Role of RAS in PI3K α Activation

Corresponding Author: Dr Dhirendra Simanshu

Version 0:

Reviewer comments:

Reviewer #1

(Remarks to the Author)

A large percentage of cancers are driven by the dysregulation of RAS GTPase and Phosphatidylinositol-3 kinase (PI3K) mediated signaling pathways. PI3K is a downstream effector of the RAS GTPase and functions as a lipid kinase to convert PIP2 into the second messenger PIP3. PIP3, in turn, stimulates multiple pathways, including mTOR signaling. A study by Murillo et al. from 2018 showed tumor regression upon disruption of the RAS/PI3K interaction. As such, understanding how these proteins interact and resolving the structure of the complex is will aid in targeting dysregulation of this pathway. The PI3K α isoform is highly mutated in human cancer and overgrowth syndromes. While a structure of PI3K γ in complex with HRAS was solved over 20 years ago, low sequence homology exists between the RAS binding domain (RBD) of PI3K α and γ isoforms. Moreover, emerging evidence suggests that in addition to RAS proteins, members of the R-Ras subfamily of small G-proteins (R-RAS, R-RAS2/TC21, and M-RAS) play key roles in PI3K α signaling and tumorigenesis, yet little is known about how KRAS and RAS-related proteins interact with PI3K α . The investigators solve X-ray structures of KRAS, RRAS2, and MRAS bound to PI3K α at relatively high resolution and evaluate residue specific interaction differences. They find that RAS interactions with PI3K α appear localized to the RBD and compare previous crystal structures of PI3K α with their RAS/PI3K α complex X-ray structures. The authors predict that conformational changes within PI3K α , upon RAS protein binding, are localized to the RBD and identify a 'glue' compound that greatly enhances complex formation. Additional comparative analyses are done to evaluate RAS interaction differences between PI3K α and γ isoforms, as well as differences in RAS binding between CRAF and PI3K α . Taken together, this work provides new molecular insights into how RAS and RAS related proteins recognize PI3K isoforms as well as the identification of a 'glue' compound that greatly enhances these interactions. Findings from this work may facilitate drug development efforts to inhibit deregulated signaling by RAS and RAS-related protein interactions with PI3K α .

This work provides new structural insights into binding interactions of RAS and RAS-related protein with PI3K α , however, the manuscript would be strengthened by the following clarifications and revisions.

1. The authors should discuss evidence that RRAS subfamily proteins activate PI3K isoforms to promote downstream signaling and tumorigenesis, as this will elevate the importance of analyzing differential binding interactions as well as how differences in binding affinity may alter downstream signaling outcomes.
2. Thermodynamic information obtained from ITC data is employed to determine reported binding affinities, yet are lacking key parameters (e.g. stoichiometry). Further, given the abundance of thermodynamic data obtained from the ITC data, the authors should correlate differential binding affinities of WT and variant RAS and RAS-related proteins with this data to provide further insight into RAS/PI3K binding interactions.
3. In Figure 2D, ITC data is used to determine binding affinity of KRAS to p110 α in the presence of a glue molecule, yet lacks sufficient data in key regions of the binding curve to extrapolate KD.
4. The authors note that X-ray structures of the complexes could only be obtained in the absence of the ABD and p85 regulatory subunit. However, Figure 2 shows the ABD domain (gray) in most panels without referencing or labeling it. Further information is needed on whether this domain was resolved or not, and whether it should be included if not present in the crystal structure.

5. The RMSD graph in Figure 4 compares a previously published PI3K α apo structure with the structure of the RRAS2/PI3K α complex crystallized by the authors. The authors should comment on whether a comparison of these structures can be made due to differences in crystallization conditions. This is exemplified by the reported RBD values ($\sim 10\text{\AA}$).

6. The apo structure of p110 α has been referenced from the PDB in other figures, but in Figure 7, no reference is provided. It is unclear whether the authors resolved the apo form structure (and if so, why it is not compared in the RMSD analysis) or whether they are deriving analyses from a previously published apo structure.

7. In Figure 6, there is insufficient evidence to claim that the orientation of the $\alpha 1$ and $\alpha 2$ helices within the RBD drive differential binding affinities of the isoforms. Mutagenesis is needed to confirm these claims.

8. It would be helpful to include a figure showing the entire binding interface of RAS and PI3K α to illustrate the overall orientation, followed by zoomed-in panels to highlight specific interactions. Key regions associated with the binding interface for both RAS and PI3K α should be labeled in all figures.

Reviewer #2

(Remarks to the Author)

Reviewer #3

(Remarks to the Author)

In this study, Czyzyk et al describe the molecular interaction of Ras family proteins with class I PI3K proteins. The authors use crystal structures of RRAS2, MRAS, and KRAS, in complex with the p110 subunit of PI3K, to illustrate commonalities and differences in binding between the different RAS families. To facilitate the crystallization of the complexes, the authors employed a point mutation in the Ras protein or the small molecule drug, D927, to increase the affinity of the proteins. ITC binding studies were performed for multiple Ras family members and for point mutations in Ras and PI3K, based on the structures, to understand disease mutations and sequence differences between family members. The experimental structures of the Ras-PI3K complex add to the Ras-PI3K structure by Pacold et al and illustrate the creative approach to achieving these structures. The analysis of the signaling differences with the different RAS is certainly unique and noteworthy. On the other hand, the abstract and conclusions overreached when stated, 'this suggests that the interaction between membrane-localized RAS and PI3K α predominantly activates PI3K α by recruiting it to the membrane.' The data provided does not support or suggest that statement.

Overall, the manuscript is well written, overall, and presents a solid body of work. Below are some comments that could help strengthen and clarify some points in the manuscript:

1. Line 46: The abstract and conclusions overreached when stated, 'this suggests that the interaction between membrane-localized RAS and PI3K α predominantly activates PI3K α by recruiting it to the membrane.'
2. Line 85: the authors list a number of ras-related family members. It could be useful to include a list of Ras proteins and their Uniprot IDs in a table for reference as well as a tree describing their positions within the Ras superfamily. Alternatively, it would be useful to include the Uniprot IDs in the methods, although this only covers proteins used in experiments, not the ones used in sequence comparisons.
3. Generally, it would be useful to have a schematic of the sequences which map the location of different regions in PI3K (e.g. r 1, r 2, RBD) and RAS (e.g. switch1, switch2, 2 strand) that are described in the manuscript.
4. Line 180: As described in the previous comment, it could be useful to comment that the adaptor binding domain spans residues 1-104, so someone who does not work on PI3K would not think it was beyond residue 1068, the other boundary of the construct listed.
5. Line 190: To facilitate crystallizations, mutations were made to improve affinity to PI3K (Q25A for KRAS, Q36A for RRAS2, and Q35A for MRAS). It is noted that the MRAS mutation had affinity comparable to wild-type but achieved 2.75A crystals. Was the resolution really due to affinity?
6. Line 196-201. The authors stated that compound D927 increased the affinity of KRAS PI3K α . This compound is in the table 1 somewhat of a view is in figure 4K. It would be a nice addition to have the chemical structure should be shown clearly as a scheme even in supplementary. Please include and remove statement "details will be published elsewhere". No relevance to this manuscript is the announcement of the GLUT4 screen.
7. Line 243. In the introduction explanation and relevance of RAS like proteins switch 1 and 2 would be desirable.
8. Line 271. In the description of the interaction of the p110 α kinase domain with RAS might be of interest to add that the residues in the area of RBD 227-232 also were observed to be part of crystal contact with the Kinase domain of a neighboring molecule in the absence of RAS. (Reference Insights into the oncogenic effects of /PIK3CA/ mutations from the structure of p110 α /p85 α (tandfonline.com))
9. Line 410: The authors use the phrase, "apo-structures of GMPPNP-bound RRAS2" which is confusing. It could be read that RRAS2 is bound to GMPPNP and the p110 does not have compound bound? In this case, it appears that apo means in the absence of PI3K. This should be restated for clarification.

Analogous to the previous comment, the titles of in Table 1 have (GMPPNP) in some columns but it is present in all samples.

10. Line 416: The authors describe state 1 and state 2 conformations of the Switch-1 region. It would be useful for the authors to explain the states.

11. Line 476: The authors should define k_5 and k_6 , and include the residue numbers (e.g. kinase alpha helix 5 (k_5)?)

12. Line 488: It would be useful to include a domain schematic diagram and residue numbers for the SH3-BH-nSH2-iSH2-cSH2.

13. Line 496: In describing k_8 - k_{11} , it would help to add residue numbers for these helices to include them in figure 6H.

14. Line 524. The section of Model of the RAS-PI3K complex on the plasma membrane, if anywhere belongs to the discussion since it is speculative. The data of the Ras-p110a new provided is homologous to the structure of ras-p110g and such models have been described before. To complete the scholarly reporting of model of the Ras-PI3K α , such models have been published before and should be referenced.

15. Line 800. The section in method "synthesis of the molecular glue compound D927" has sentences that don't seem to belong to methods or are not understood as written.

Comments on Figures (using the lines in figure captions)

16. Figure 1 would be more compelling if all the panels have the same range of ΔH kcal/mol (axis Y) to clearly display strong binding to weak to no binding.

17. Add in the columns of Table 1 the corresponding PDB ID

Reviewer #4

(Remarks to the Author)

Version 1:

Reviewer comments:

Reviewer #1

(Remarks to the Author)

1. The authors should discuss evidence that RRAS subfamily proteins activate PI3K isoforms to promote downstream signaling and tumorigenesis, as this will elevate the importance of analyzing differential binding interactions as well as how differences in binding affinity may alter downstream signaling outcomes.

Reply: We appreciate the suggestion to enhance the discussion. We have now added additional text (lines 518-533) discussing the evidence that RRAS subfamily proteins activate PI3K isoforms, contributing to downstream signaling and tumorigenesis. This discussion highlights the significance of differential binding interactions and how variations in binding affinity may influence downstream signaling outcomes.

Reviewer response. While the authors effectively highlight the importance of RRAS2 in PI3K α activation in the discussion, emphasizing significant work from the Bustelo group and others that demonstrate the dependence of PI3K α signaling on RRAS2, the manuscript would benefit from addition of text in the introduction to set the stage for why structural studies of RRAS sub-family members are employed.

2. Thermodynamic information obtained from ITC data is employed to determine reported binding affinities, yet are lacking key parameters (e.g. stoichiometry). Further, given the abundance of thermodynamic data obtained from the ITC data, the authors should correlate differential binding affinities of WT and variant RAS and RAS-related proteins with this data to provide further insight into RAS/PI3K binding interactions.

Reply: Thank you for the feedback. We have now included stoichiometry (N-value) in the ITC data (Supplementary Tables 1, 2 and 3) used to determine reported binding affinities. We have also expanded our analysis to include a correlation between the thermodynamic data obtained from ITC and the differential binding affinities of WT and variant RAS and RAS-related proteins.

Reviewer response. The authors have now incorporated our suggestion to include stoichiometry (N-value) in ITC analyses, with most values falling within the acceptable range (though one sample had an N of 0.75). They also address the differences in ΔH and $-\Delta S$, shedding light on thermodynamics associated with the interactions.

3. In Figure 2D, ITC data is used to determine binding affinity of KRAS to p110 α in the presence of a glue molecule, yet lacks sufficient data in key regions of the binding curve to extrapolate KD.

Reply: We acknowledge the limitations in the ITC data presented in Figure 2D and have addressed this by conducting additional ITC experiments with p110 α and KRAS in the presence of D927. These results are now included in the updated Figure 2E and Supplementary Fig. 5A, B). We used a smaller injection volume and increased the number of injections to

provide a more detailed binding curve, ensuring accurate K_D extrapolation.

Reviewer response. The authors have conducted additional ITC experiments with the glue compound, including additional measurements in key regions of the binding curve, enabling better extrapolation of the K_D (now 49 nM compared to 34 nM reported previously).

4. The authors note that X-ray structures of the complexes could only be obtained in the absence of the ABD and p85 regulatory subunit. However, Figure 2 shows the ABD domain (gray) in most panels without referencing or labeling it. Further information is needed on whether this domain was resolved or not, and whether it should be included if not present in the crystal structure.

Reply: We have clarified in the text that the ABD domain, now shown in white instead of gray, was not resolved in the crystal structure. The gray regions in various panels of Figure 2 represent the linker regions between different domains. To prevent confusion between the ABD domain and linker regions, we have updated Figure 2A to depict the missing regions in white and the linker regions in gray. The same color scheme has been consistently applied across all panels of Figure 2 to avoid any confusion.

Reviewer response. This issue was not addressed. White/gray colored helices and their respective surface structures remain unlabeled in Figure 2 (Panels C, D and F) and not mentioned in the figure legend. Additionally, it is hard to distinguish from the structures what is white versus gray.

5. The RMSD graph in Figure 4 compares a previously published PI3K α apo structure with the structure of the RRAS2/PI3K α complex crystallized by the authors. The authors should comment on whether a comparison of these structures can be made due to differences in crystallization conditions. This is exemplified by the reported RBD values ($\sim 10\text{\AA}$).

Reply: Yes, such a comparison can be made despite the structures being obtained under different crystallization conditions. However, it is important to determine whether the observed differences are due to crystal contacts or are genuinely caused by protein-protein interactions. We have analyzed crystal contacts in both the apo and complex structures, particularly in regions where we observe significant conformational changes, and have commented on whether these changes are attributable to crystal contacts or RAS-p110 α interactions.

Reviewer response. The authors state that that conformational changes are due to crystal contacts in lines 202 and 203 of the revised manuscript, but additional discussion would be helpful regarding how comparisons can be drawn from structures under different crystallization conditions and how different the crystallization conditions are. It would also be helpful to highlight residues/regions missing from either structure that prevent comparative analyses and provide additional details on the type of analysis conducted. Did the authors quantify contact areas or assess B-factors in the regions of interest? Additionally, since the RBD region shows a $\sim 10\text{\AA}$ deviation, can the authors clarify if the analysis suggests these differences are likely due to crystal contacts or represent actual structural changes induced by RRAS2 binding?

6. The apo structure of p110 α has been referenced from the PDB in other figures, but in Figure 7, no reference is provided. It is unclear whether the authors resolved the apo form structure (and if so, why it is not compared in the RMSD analysis) or whether they are deriving analyses from a previously published apo structure.

Reply: This was an oversight on our part, and we have now included the appropriate reference for the apo structure of p110 α used in Figure 7. We have clarified in the text that the apo structure was obtained from a previously published work, and this published apo structure was used for the RMSD analysis.

Reviewer response. The authors have now fixed these errors.

7. In Figure 6, there is insufficient evidence to claim that the orientation of the $\alpha 1$ and $\alpha 2$ helices within the RBD drive differential binding affinities of the isoforms. Mutagenesis is needed to confirm these claims.

Reply: Sorry for the lack of clarity in this section. We did not intend to imply that the orientation of the $\alpha 1$ and $\alpha 2$ helices within the RBD drives the differential binding affinities of class I PI3K isoforms with RAS proteins. Instead, we highlighted structure-based sequence alignments showing a lack of conservation of RAS-interacting residues and the deletion of multiple residues between the $\beta 1$ - $\beta 2$ strands and within the $\alpha 2$ -helix in PI3K β , PI3K γ , and PI3K δ compared to PI3K α , which likely explains these differences in binding affinities. Given the multiple deletions and differences in RAS-interacting residues among the other PI3K isoforms, it is not feasible to conduct mutagenesis studies to fully explain the differential binding affinities. We have reworded this section to reflect these points more clearly.

Reviewer response. The authors have now clarified their conclusion from the figures and have included lines 443-445, highlighting the differential interactions and not the orientations of the helices.

8. It would be helpful to include a figure showing the entire binding interface of RAS and PI3K α to illustrate the overall orientation, followed by zoomed-in panels to highlight specific interactions. Key regions associated with the binding interface for both RAS and PI3K α should be labeled in all figures.

Reply: As suggested, we have updated Figure 3 to provide a comprehensive view of the entire binding interface between RAS and PI3K α . The updated Figure 3 now includes zoomed-in panels (Fig. 3B, 3C, and 3D) highlighting specific interactions, with key regions of the binding interface clearly labeled for clarity.

Reviewer response. The revised Figure 3 better depicts the protein-protein interface.

Reviewer #2

(Remarks to the Author)

Reviewer #3

(Remarks to the Author)

The authors have addressed the comments raised

Reviewer #4

(Remarks to the Author)

Point-by-point response to reviewer's comments

We thank the reviewers for their detailed and thoughtful review of our manuscript. We greatly appreciate the timely processing of our submission and the reviewers' positive feedback and insightful suggestions. In the revised manuscript, we have addressed the concerns of all reviewers, and we believe that their feedback has significantly enhanced the quality of our manuscript. We also shortened the manuscript to align with the guidelines of Nature Communications and enhanced all main figures for improved clarity and presentation. In the sections that follow, we have provided point-by-point responses (in blue) to the reviewers' comments.

Reviewer #1:

A large percentage of cancers are driven by the dysregulation of RAS GTPase and Phosphatidylinositol-3 kinase (PI3K) mediated signaling pathways. PI3K is a downstream effector of the RAS GTPase and functions as a lipid kinase to convert PIP2 into the second messenger PIP3. PIP3, in turn, stimulates multiple pathways, including mTOR signaling. A study by Murillo et al. from 2018 showed tumor regression upon disruption of the RAS/PI3K interaction. As such, understanding how these proteins interact and resolving the structure of the complex is will aid in targeting dysregulation of this pathway.

The PI3 α isoform is highly mutated in human cancer and overgrowth syndromes. While a structure of PI3K γ in complex with HRAS was solved over 20 years ago, low sequence homology exists between the RAS binding domain (RBD) of PI3K α and γ isoforms. Moreover, emerging evidence suggests that in addition to RAS proteins, members of the R-Ras subfamily of small G-proteins (R-RAS, R-RAS2/TC21, and M-RAS) play key roles in PI3K α signaling and tumorigenesis, yet little is known about how KRAS and RAS-related proteins interact with PI3K α .

The investigators solve X-ray structures of KRAS, RRAS2, and MRAS bound to PI3K α at relatively high resolution and evaluate residue specific interaction differences. They find that RAS interactions with PI3K α appear localized to the RBD and compare previous crystal structures of PI3K α with their RAS/PI3K α complex X-ray structures. The authors predict that conformational changes within PI3K α , upon RAS protein binding, are localized to the RBD and identify a 'glue' compound that greatly enhances complex formation. Additional comparative analyses are done to evaluate RAS interaction differences between PI3K α and γ isoforms, as well as differences in RAS binding between CRAF and PI3K α . Taken together, this work provides new molecular insights into how RAS and RAS related proteins recognize PI3K isoforms as well as the identification of a 'glue' compound that greatly enhances these interactions. Findings from this

work may facilitate drug development efforts to inhibit deregulated signaling by RAS and RAS-related protein interactions with PI3K α .

This work provides new structural insights into binding interactions of RAS and RAS-related protein with PI3K α , however, the manuscript would be strengthened by the following clarifications and revisions.

1. The authors should discuss evidence that RRAS subfamily proteins activate PI3K isoforms to promote downstream signaling and tumorigenesis, as this will elevate the importance of analyzing differential binding interactions as well as how differences in binding affinity may alter downstream signaling outcomes.

Reply: We appreciate the suggestion to enhance the discussion. We have now added additional text (lines 518-533) discussing the evidence that RRAS subfamily proteins activate PI3K isoforms, contributing to downstream signaling and tumorigenesis. This discussion highlights the significance of differential binding interactions and how variations in binding affinity may influence downstream signaling outcomes.

2. Thermodynamic information obtained from ITC data is employed to determine reported binding affinities, yet are lacking key parameters (e.g. stoichiometry). Further, given the abundance of thermodynamic data obtained from the ITC data, the authors should correlate differential binding affinities of WT and variant RAS and RAS-related proteins with this data to provide further insight into RAS/PI3K binding interactions.

Reply: Thank you for the feedback. We have now included stoichiometry (N-value) in the ITC data (Supplementary Tables 1, 2 and 3) used to determine reported binding affinities. We have also expanded our analysis to include a correlation between the thermodynamic data obtained from ITC and the differential binding affinities of WT and variant RAS and RAS-related proteins.

3. In Figure 2D, ITC data is used to determine binding affinity of KRAS to p110 α in the presence of a glue molecule, yet lacks sufficient data in key regions of the binding curve to extrapolate K_D .

Reply: We acknowledge the limitations in the ITC data presented in Figure 2D and have addressed this by conducting additional ITC experiments with p110 α and KRAS in the presence of D927. These results are now included in the updated Figure 2E and Supplementary Fig. 5A, B). We used a smaller injection volume and increased the number of injections to provide a more detailed binding curve, ensuring accurate K_D extrapolation.

4. The authors note that X-ray structures of the complexes could only be obtained in the absence of the ABD and p85 regulatory subunit. However, Figure 2 shows the ABD domain (gray) in most panels without referencing or labeling it. Further information is needed on whether this domain was resolved or not, and whether it should be included if not present in the crystal structure.

Reply: We have clarified in the text that the ABD domain, now shown in white instead of gray, was not resolved in the crystal structure. The gray regions in various panels of Figure 2 represent the linker regions between different domains. To prevent confusion between the ABD domain and linker regions, we have updated Figure 2A to depict the missing regions in white and the linker regions in gray. The same color scheme has been consistently applied across all panels of Figure 2 to avoid any confusion.

5. The RMSD graph in Figure 4 compares a previously published PI3K α apo structure with the structure of the RRAS2/PI3K α complex crystallized by the authors. The authors should comment on whether a comparison of these structures can be made due to differences in crystallization conditions. This is exemplified by the reported RBD values ($\sim 10\text{\AA}$).

Reply: Yes, such a comparison can be made despite the structures being obtained under different crystallization conditions. However, it is important to determine whether the observed differences are due to crystal contacts or are genuinely caused by protein-protein interactions. We have analyzed crystal contacts in both the apo and complex structures, particularly in regions where we observe significant conformational changes, and have commented on whether these changes are attributable to crystal contacts or RAS-p110 α interactions.

6. The apo structure of p110 α has been referenced from the PDB in other figures, but in Figure 7, no reference is provided. It is unclear whether the authors resolved the apo form structure (and if so, why it is not compared in the RMSD analysis) or whether they are deriving analyses from a previously published apo structure.

Reply: This was an oversight on our part, and we have now included the appropriate reference for the apo structure of p110 α used in Figure 7. We have clarified in the text that the apo structure was obtained from a previously published work, and this published apo structure was used for the RMSD analysis.

7. In Figure 6, there is insufficient evidence to claim that the orientation of the $\alpha 1$ and $\alpha 2$ helices within the RBD drive differential binding affinities of the isoforms. Mutagenesis is needed to confirm these claims.

Reply: Sorry for the lack of clarity in this section. We did not intend to imply that the orientation of the $\alpha 1$ and $\alpha 2$ helices within the RBD drives the differential binding affinities of class I PI3K isoforms with RAS proteins. Instead, we highlighted structure-based sequence alignments showing a lack of conservation of RAS-interacting residues and the deletion of multiple residues between the $r\beta 1$ - $r\beta 2$ strands and within the $\alpha 2$ -helix in PI3K β , PI3K γ , and PI3K δ compared to PI3K α , which likely explains these differences in binding affinities. Given the multiple deletions and differences in RAS-interacting residues among the other PI3K isoforms, it is not feasible to conduct mutagenesis studies to fully explain the differential binding affinities. We have reworded this section to reflect these points more clearly.

8. It would be helpful to include a figure showing the entire binding interface of RAS and PI3K α to illustrate the overall orientation, followed by zoomed-in panels to highlight specific interactions. Key regions associated with the binding interface for both RAS and PI3K α should be labeled in all figures.

Reply: As suggested, we have updated Figure 3 to provide a comprehensive view of the entire binding interface between RAS and PI3K α . The updated Figure 3 now includes zoomed-in panels (Fig. 3B, 3C, and 3D) highlighting specific interactions, with key regions of the binding interface clearly labeled for clarity.

Reviewer #2:

Reply: We appreciate the feedback and commend the Nature Communications initiative for supporting the training and recognition of Early Career Researchers through co-reviewing manuscripts.

Reviewer #3:

In this study, Czyzyk et al describe the molecular interaction of Ras family proteins with class I PI3K proteins. The authors use crystal structures of RRAS2, MRAS, and KRAS, in complex with the p110 subunit of PI3K, to illustrate commonalities and differences in binding between the

different RAS families. To facilitate the crystallization of the complexes, the authors employed a point mutation in the Ras protein or the small molecule drug, D927, to increase the affinity of the proteins. ITC binding studies were performed for multiple Ras family members and for point mutations in Ras and PI3K, based on the structures, to understand disease mutations and sequence differences between family members. The experimental structures of the Ras-PI3K complex add to the Ras-PI3K structure by Pacold et al and illustrate the creative approach to achieving these structures. The analysis of the signaling differences with the different RAS is certainly unique and noteworthy. On the other hand, the abstract and conclusions overreached when stated, 'this suggests that the interaction between membrane-localized RAS and PI3Ka predominantly activates PI3Ka by recruiting it to the membrane.' The data provided does not support or suggest that statement.

Overall, the manuscript is well written, overall, and presents a solid body of work. Below are some comments that could help strengthen and clarify some points in the manuscript:

1. Line 46: The abstract and conclusions overreached when stated, 'this suggests that the interaction between membrane-localized RAS and PI3Ka predominantly activates PI3Ka by recruiting it to the membrane.'

Reply: We appreciate the feedback. As suggested, we have amended the abstract and conclusions to more precisely reflect our findings, eliminating any overreaching claims. We have removed these details from the abstract and modified the text in the discussion to avoid making any claims about it.

2. Line 85: the authors list a number of ras-related family members. It could be useful to include a list of Ras proteins and their Uniprot IDs in a table for reference as well as a tree describing their positions within the Ras superfamily. Alternatively, it would be useful to include the Uniprot IDs in the methods, although this only covers proteins used in experiments, not the ones used in sequence comparisons.

Reply: Thank you for this suggestion. We have included a new Supplementary Fig 13 listing the RAS proteins used in Figure 4A and their UniProt IDs, along with a phylogenetic tree that describes their positions within the RAS family. Additionally, we have added UniProt IDs in the methods section for the proteins used in experiments.

3. Generally, it would be useful to have a schematic of the sequences which map the location of different regions in PI3K (e.g. r1, r2, RBD) and RAS (e.g. switch1, switch2, 2 strand) that are described in the manuscript.

Reply: We have included a new Supplementary Figure 6 that provides a schematic mapping of the different regions in PI3K (e.g., r1, r2, RBD) and RAS (e.g., switch 1, switch 2, \$\beta\$ 2-strand) to enhance clarity.

4. Line 180: As described in the previous comment, it could be useful to comment that the adaptor binding domain spans residues 1-104, so someone who does not work on PI3K would not think it was beyond residue 1068, the other boundary of the construct listed.

Reply: We have included additional details clarifying that the adaptor binding domain spans residues 1-104, ensuring that readers unfamiliar with PI3K do not mistakenly associate it with regions beyond residue 1068.

5. Line 190: To facilitate crystallizations, mutations were made to improve affinity to PI3K (Q25A for KRAS, Q36A for RRAS2, and Q35A for apo). It is noted that the MRAS mutation had affinity comparable to wild-type but achieved 2.75Å crystals. Was the resolution really due to affinity?

Reply: As we described, the wild-type and Q35A mutant of MRAS exhibited a similar affinity for p110 \$\alpha\$, suggesting that the improved resolution (2.75Å) for the MRAS-p110 \$\alpha\$ complex was not solely due to affinity. Other factors like crystallization conditions and lattice contacts likely contributed to the enhanced crystal quality. We have clarified this in the revised manuscript to avoid implying that affinity was the only factor responsible for the resolution.

6. Line 196-201. The authors stated that compound D927 increased the affinity of KRAS PI3K α . This compound is in the table 1 somewhat of a view is in figure 4K. It would be a nice addition to have the chemical structure should be shown clearly as a scheme even in supplementary. Please include and remove statement "details will be published elsewhere". No relevance to this manuscript is the announcement of the GLUT4 screen.

Reply: We have included a schematic representation of the interaction between compound D927 and p110 \$\alpha\$ /KRAS, generated using Ligplot+, in the Supplementary Fig. 5. Additionally, we have removed the statement 'details will be published elsewhere' to keep the focus on the relevant findings and have also eliminated the mention of the GLUT4 screen to keep the manuscript focused on the current study.

7. Line 243. In the introduction, explanation and relevance of RAS, like proteins switch 1 and 2, would be desirable.

Reply: The introduction has been revised to include details on switch-I and switch-II, providing context for their role in the RAS-p110 α (RBD) interaction.

8. Line 271. In the description of the interaction of the p110a kinase domain with RAS might be of interest to add that the residues in the area of RBD 227-232 also were observed to be part of crystal contact with the Kinase domain of a neighboring molecule in the absence of RAS.

Reply: Thank you for the suggestion. We have now included the observation that residues in the RBD 227-232 region were involved in crystal contact with the kinase domain of a neighboring molecule in the absence of RAS, as supported by the referenced study.

9. Line 410: The authors use the phrase, “apo-structures of GMPPNP-bound RRAS2” which is confusing. It could be read that RRAS2 is bound to GMPPNP and the p110 does not have compound bound? In this case, it appears that apo means in the absence of PI3K. This should be restated for clarification. Analogous to the previous comment, the titles of in Table 1 have (GMPPNP) in some columns but it is present in all samples.

Reply: We have revised the text to clarify that 'apo' refers to the absence of PI3K, not the absence of GMPPNP. Additionally, we have corrected the titles in Table 1 to ensure consistency, indicating that GMPPNP is present in all samples.

10. Line 416: The authors describe state 1 and state 2 conformations of the Switch-1 region. It would be useful for the authors to explain the states.

Reply: We have now included the details of the state 1 and state 2 conformations of the Switch-I region, emphasizing their structural distinctions to enhance clarity.

11. Line 476: The authors should define k5 and k6, and include the residue numbers (e.g kinase alpha helix 5 (k5)?

Reply: We have now defined helices K α 5 and K α 6 and the specific residue numbers in the text.

12. Line 488: It would be useful to include a domain schematic diagram and residue numbers for the SH3-BH-nSH2-iSH2-cSH2.

Reply: As suggested, a schematic diagram of the domain structure of p85, including residue numbers, has been incorporated into Fig. 2A of the manuscript.

13. Line 496: In describing k8-k11, it would help to add residue numbers for these helices to include them in figure 6H.

Reply: Thank you. To enhance the clarity of Fig. 6H, we have now included the residue numbers for helices K α 8-K α 11 in the description.

14. Line 524. The section of Model of the RAS-PI3K complex on the plasma membrane, if anywhere belongs to the discussion since it is speculative. The data of the Ras-p110a new provided is homologous to the structure of ras-p110g and such models have been described before. To complete the scholarly reporting of model of the Ras-PI3Ka, such models have been published before and should be referenced.

Reply: Thank you for your feedback. As suggested, we have moved the section on the "Model of the RAS-PI3K complex on the plasma membrane" to the Discussion. We have also incorporated references to previous reports describing similar models to provide a more comprehensive and scholarly context for our model.

15. Line 800. The section in the method "synthesis of the molecular glue compound D927" has sentences that don't seem to belong to methods or are not understood as written.

Reply: Thank you for pointing that out. We have revised the section on the "synthesis of the molecular glue compound D927" to improve clarity and coherence. Additionally, we have referenced the previous work where the method is described in detail to provide more context.

Comments on Figures (using the lines in figure captions)

16. Figure 1 would be more compelling if all the panels had the same range of DeltaH kcal/mol (axis Y) to clearly display strong binding to weak to no binding.

Reply: We appreciate this suggestion and have revised Figure 1 to ensure different panels have the same DeltaH kcal/mol range on the Y-axis. We agree that this adjustment makes displaying the range from strong binding to weak or no binding easier.

17. Add in the columns of Table 1 the corresponding PDB ID.

Reply: As suggested, we have now included the corresponding PDB ID in Table 1

Reviewer #4:

Reply: Thank you for the feedback. We recognize and value the efforts of the Nature Communications initiative to promote the training and recognition of early career researchers through the co-reviewing process.

Point-by-point response to Reviewer #1 comments:

1. While the authors effectively highlight the importance of RRAS2 in PI3K α activation in the discussion, emphasizing significant work from the Bustelo group and others that demonstrate the dependence of PI3K α signaling on RRAS2, the manuscript would benefit from addition of text in the Introduction to set the stage for why structural studies of RRAS sub-family members are employed.

Reply: We appreciate the reviewer's feedback. We have included text in the Introduction to provide context on the role of RRAS subfamily members in PI3K activation and tumorigenesis. This emphasizes the significance of studying the structural and functional aspects of RRAS2 and related proteins, setting the stage for the detailed analysis of their interactions with PI3K isoforms presented in the manuscript.

2. The authors state that conformational changes are due to crystal contacts in lines 202 and 203 of the revised manuscript, but additional discussion would be helpful regarding how comparisons can be drawn from structures under different crystallization conditions and how different the crystallization conditions are. It would also be helpful to highlight residues/regions missing from either structure that prevent comparative analyses and provide additional details on the type of analysis conducted. Did the authors quantify contact areas or assess B-factors in the regions of interest? Additionally, since the RBD region shows a ~ 10 Å deviation, can the authors clarify if the analysis suggests these differences are likely due to crystal contacts or represent actual structural changes induced by RRAS2 binding?

Reply: We appreciate the reviewer's suggestion for additional discussion regarding the potential impact of crystal contacts on the observed conformational changes. As noted in the manuscript, the RBD region (residues 233-246), previously disordered in apo-PI3K α structures lacking crystal contacts, is well-defined in all three structures presented here. This stabilization is attributed to interactions with RRAS2/MRAS/KRAS rather than artifacts caused by crystal contacts.

While crystal contacts may influence protein conformation, the consistent observation of this region's stabilization across multiple crystal forms, particularly in the presence of RRAS2/MRAS/KRAS, strongly suggests that the observed conformational change is biologically relevant. Regarding the ~ 10 Å deviation in the RBD, we have clarified that this is primarily due to structural stabilization induced by RRAS2/MRAS/KRAS binding and not an artifact of crystal contacts.

Although a detailed analysis of crystal contact areas, B-factors, and missing residues could provide additional insights, we believe such details would exceed the scope and word limit of the manuscript. The current analysis focuses on the key structural changes induced by RAS binding and provides a concise and clear understanding of their molecular implications.

3. White/gray colored helices and their respective surface structures remain unlabeled in Figure 2 (Panels C, D and F) and not mentioned in the figure legend. Additionally, it is hard to distinguish from the structures what is white versus gray.

Reply: We appreciate the reviewer's feedback regarding the labeling and representation of the ABD domain and linker regions in Figure 2. To clarify, the missing ABD domain, depicted in white in Figure 2A, is not shown in any of the structural figure panels (C, D, and F) in Figure 2. Only the linker regions, represented in gray, are included in these panels. To address potential confusion, we have now labeled the gray regions in all panels of Figure 2 to explicitly indicate the linker regions. Additionally, we have updated the figure legend to provide a detailed explanation of the color scheme, specifying that the ABD domain shown in white in Figure 2A is absent from panels C, D, and F. We believe these changes enhance the clarity and visual interpretation of Figure 2, effectively addressing the reviewer's concerns.